# From Abstraction to Instantiation: Learning Behavioral Representation for Vision-Language-Action Model

**Bing Hu** [1]  **Zaijing Li** [1 2]  **Rui Shao** [† 1 2 3]  **Junda Chen** [1]  **April Hua Liu** [4]  **Wei-Shi Zheng** [3 5]  **Liqiang Nie** [1]

BehaviorVLA.github.io

## Abstract

Vision-Language-Action (VLA) models often suffer from performance degradation under distribution shifts, as they struggle to learn generalized behavior representations across varying environments. While existing approaches attempt to construct behavior representations through action-centric latent variables, they are often limited by short-horizon temporal fragmentation and static execution-alignment, leading to inconsistent behaviors in complex scenarios. To address these limitations, we propose **BehaviorVLA**, a framework that facilitates robust manipulation through the learning of a temporally coherent behavioral representations. Our approach features two symmetric components: (1) the **Visuomotor Behavior Encoder (VBE)**, which utilizes a causal Mamba-based architecture to aggregate long-horizon trajectory information into a unified behavior representation; and (2) the **Phase-conditioned Behavior Decoder (PBD)**, which decodes this representation into precise actions by dynamically aligning task-level priors with real-time execution progress. Experiments on RoboTwin 2.0, LIBERO, and CALVIN demonstrate state-of-the-art success rates of 58%, 98%, and 4.36 (Avg.Len), respectively. Notably, in real-world sim-to-real transfer, BehaviorVLA matches the performance of OpenVLA-OFT using only 50% of the demonstration data, showcasing its superior data efficiency and generalization.

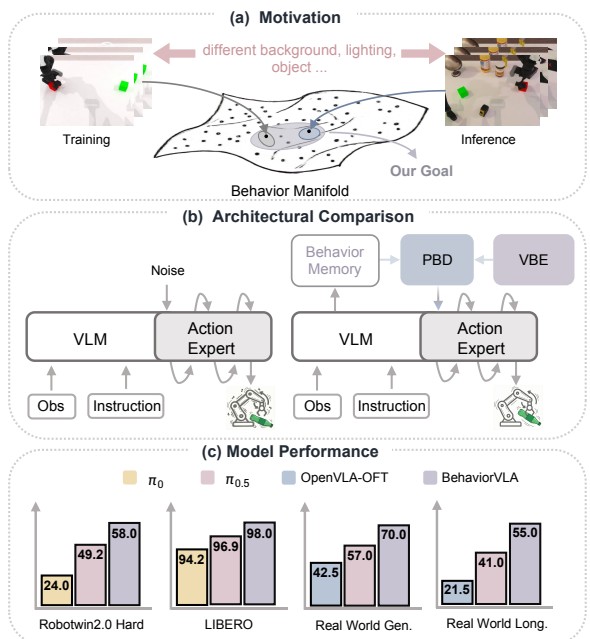

*Figure 1.* (a) **Motivation**: Standard VLAs learn mappings in high-dimensional space without explicit manifold constraints. In contrast, our goal is to learn a low-dimensional behavioral manifold to capture transferable patterns. (b) **Architecture**: Unlike standard VLAs, BehaviorVLA incorporates the Visuomotor Behavior Encoder (VBE), Phase-conditioned Behavior Decoder (PBD), and Behavior Memory Bank to learn and retrieve these latent behaviors. (c) **Performance**: Our model consistently outperforms baselines across Robotwin 2.0, LIBERO, and Real-World.

## 1. Introduction

Recent advances in Vision-Language-Action (VLA) models (Kim et al., 2024; Shi et al., 2025; Bu et al., 2025; Qu et al., 2025; Black et al., 2024; Intelligence et al., 2025) have demonstrated impressive capabilities in robotic manipulation. These models are typically trained on large scale simulation datasets (Liu et al., 2023; Mees et al., 2022; Chen et al., 2025b) and leverage a vision-language backbone (Liu et al., 2024a) to map observations and instructions directly into action sequences. However, a critical challenge remains: VLA performance often degrades substantially under distribution shifts (Kim et al., 2024; Black et al., 2024), especially

---

[†]Corresponding Author. [1]Harbin Institute of Technology, Shenzhen [2]PengCheng Laboratory [3]Shenzhen Loop Area Institute [4]Shanghai University of Finance and Economics [5]Sun Yat-sen University. Correspondence to: Rui Shao <shaorui@hit.edu.cn>.

when transferring from simulation to the real world (Aljalbout et al., 2025). For instance, a VLA model trained in simulation may fail catastrophically in the real world due to variations in object material, scene clutter, or camera viewpoint. Such pronounced brittleness suggests that current models often overfit to training distributions rather than capturing the generalized behavior representations.

As illustrated in Figure 1, from the perspective of the manifold hypothesis (Fefferman et al., 2016; Narayanan & Mitter, 2010), high dimensional visuomotor trajectories concentrate near a low-dimensional manifold embedded in the ambient space. However, standard VLA models (Intelligence et al., 2025; Kim et al., 2024) learn their mappings directly in the ambient high dimensional space without explicit manifold constraints. Consequently, under domain shifts, predicted actions often drift away from the valid task manifold, leading to invalid or suboptimal behaviors. While extensive real-world fine-tuning can mitigate this (Lin et al., 2025; Bjorck et al., 2025), it is prohibitively expensive and hard to scale. A more principled approach is to learn a compact latent space that captures transferable behavioral patterns, projecting high-dimensional observations onto a lower-dimensional behavioral manifold.

Prior works have attempted to learn such latent action space via Variational Autoencoder (VAE). Skill-based approaches, such as BeT (Shafiullah et al., 2022) and VQ-BeT (Lee et al., 2024), utilize Vector Quantization (VQ) to discretize actions into latent codes, while Action Chunking Transformer (ACT) (Zhao et al., 2023b) aggregates actions into temporal chunks. While effective for modeling local smoothness, these methods face two fundamental limitations. (i) **Short-horizon temporal fragmentation**. By slicing trajectories into independent chunks or discrete codes, they often fail to capture the long-term dependencies essential for complex manipulation, resulting in a lack of global coherence. (ii) **Static execution-alignment**. These models typically decode actions from static latent variables without accounting for real-time execution progress. This disconnect often leads to temporal misalignment, where the generated action sequence drifts from the actual state of the environment.

To bridge this gap, we introduce a novel framework centered on learning visuomotor behavior representations. We posit that a robust VLA model requires two distinct capabilities: (1) a *specific-to-general* abstraction that distills diverse demonstrations into a unified behavior representation, and (2) a *general-to-specific* instantiation that projects this abstract behavior into precise, situation-aware actions.

To achieve specific-to-general abstraction, we propose the **V**isuomotor **B**ehavior **E**ncoder (**VBE**). Unlike standard encoders that process frames independently, VBE features a causal three-stream architecture designed to capture full trajectory dynamics. Temporally, it utilizes the Mamba framework to independently model the evolution of vision, action, and behavior streams, ensuring global consistency across the entire horizon. Spatially, at each timestep, it employs a cross-attention mechanism to fuse visual and action information into the behavior token. This design ensures that the learned representation captures the persistent logic of the task rather than transient environmental noise, addressing the issue of temporal fragmentation.

For general-to-specific instantiation, we propose the **P**hase-conditioned **B**ehavior **D**ecoder (**PBD**), which follows a Predictor-Corrector paradigm to map behavior representations into precise actions. In the prediction stage, PBD unfolds the behavior representation into a sequence of structural priors. By incorporating a learned phase state and Progress-Attention, the decoder dynamically aligns these priors with the real-time execution progress, effectively mitigating the temporal drift common in static models. In the correction stage, these phase-aligned priors guide the vector field of a Conditional Flow policy. This hierarchical design integrates the stability of global task structures with the precision of local reactive control, enabling consistent execution of complex manipulation.

Building upon these modules, we present **BehaviorVLA**, a unified VLA model that integrates the vision-language backbone with VBE and PBD. We conduct extensive experiments across both simulation benchmarks and real-world scenarios. In simulation, BehaviorVLA achieves superior performance, recording success rate of 98%, 4.36, and 58% on LIBERO (Liu et al., 2023), CALVIN (Mees et al., 2022), and RoboTwin 2.0 (Chen et al., 2025b), respectively. In the real world, across 8 challenging manipulation tasks, our model outperforms strong baselines with an average success rate improvement of 63%. Notably, BehaviorVLA demonstrates exceptional data efficiency in sim-to-real adaptation, matching the performance of the fully fine-tuned OpenVLA-OFT (Kim et al., 2025) while utilizing only 50% of the Real-World demostrations.

In summary, our contributions are threefold:

- We propose the **Visuomotor Behavior Encoder (VBE)**, a causal three-stream architecture that aggregates long-horizon trajectory dynamics into unified behavior representations, effectively capturing persistent task logic while filtering environmental noise.

- We introduce the **Phase-conditioned Behavior Decoder (PBD)**, which instantiates behavior representations through a Predictor-Corrector paradigm. By incorporating a learned phase state, PBD ensures strict synchronization between structural priors and real-time execution progress.

- We develop **BehaviorVLA**, a unified framework that achieves superior performance on both simulation benchmarks and Real-World evaluation.

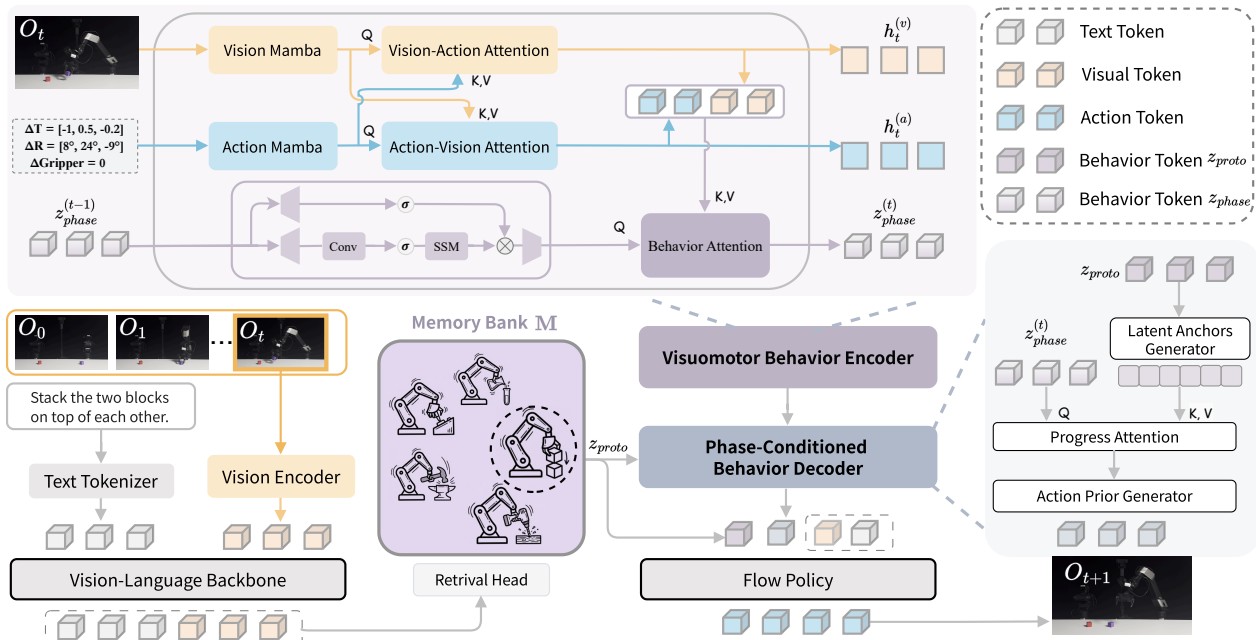

*Figure 2.* Overview of BehaviorVLA. Given an instruction and observation, the Vision-Language backbone first integrates multimodal information to retrieve a global prototype $z_{proto}$ from the Memory Bank. The retrieval is performed only once at the beginning of each episode, and the retrieved prototype remains fixed during execution as a stable behaviorial prior. Simultaneously, the Visuomotor Behavior Encoder models the current phase state $z_{phase}$ in an online fashion. Both representations are then fused by the Phase-Conditioned Behavior Decoder and Flow Policy to decode the final action. Specifically, the VBE features a three-stream architecture (Vision, Action, and Behavior) that utilizes Mamba to dynamically model the temporal evolution of each sequence, while employing cross-attention mechanisms to aggregate visual and action information into the current $z_{phase}$.

## 2. Related Work

**VLA Models and Retrieval Mechanisms.** Recent Vision-Language-Action (VLA) models have demonstrated impressive capabilities by scaling up simulation data (Kim et al., 2024; Zitkovich et al., 2023; Black et al., 2025) or integrating retrieval-based memory systems (Shi et al., 2025; Li et al., 2025b; Lin et al., 2026) to handle partial observability. However, standard VLAs often rely on implicit encodings that fail to capture the structural isomorphism of manipulation tasks, while existing retrieval methods typically treat retrieved trajectories as static contexts without explicit execution progress tracking. This often leads to geometric fragility and phase misalignment during sim-to-real transfer. To address these limitations, our **Visuomotor Behavior Encoder (VBE)** explicitly disentangles behavior into time-invariant prototypes and time-variant phase latents, ensuring both structural consistency and precise progress alignment.

**Generative Policies in Robotics.** To mitigate mode averaging in imitation learning (An et al., 2025), generative policies like diffusion policy (Chi et al., 2025a) and flow policy (Rouxel et al., 2024) have become the standard for robot control. Recent VLAs(Black et al., 2025; Black et al.; Shukor et al., 2025) extend this via hierarchical VLM-action expert architectures. However, relying on uninformative pri-

ors poses fundamental limitations in contact-rich tasks: (1) lack of progress awareness: stochastic sampling without phase constraints causes latent stage jumping and temporal inconsistency. (2) contact instability: iterative generation noise induces high-frequency jitter and premature actuation. To address this, we introduce a **Phase-Conditioned Prior Decoder(PBD)** that generates a phase-consistent prior as the predictor for structural guidance, coupled with a flow policy as the corrector for residual dynamics.

**History-based and in-context policies.** Recent robot policies increasingly exploit long-horizon history to improve action generation. RPT (Radosavovic et al., 2023) performs sensorimotor pre-training with masked prediction over image, state, and action tokens, while ICRT (Fu et al., 2025) formulates in-context imitation as next-token prediction for training-free adaptation. Recent works further adopt sequence-efficient architectures: MTIL (Zhou et al., 2025) encodes full trajectory history with Mamba, and RoboSSM (Yoo et al., 2025) uses state-space models for efficient long-context imitation. Different from these methods that mainly treat history as contextual input for direct action prediction, BehaviorVLA factorizes history into a retrieved global prototype and an online phase state, thereby providing both structural guidance and progress-aware alignment for the downstream flow policy.

# 3. Method

In this section, we first give an overview of our proposed BehaviorVLA (as depicted in Figure 2). Next, we introduce the details of **V**isuomotor **B**ehavior **E**ncode in Sec. 3.2. Subsequently, we elaborate on how to implement the proposed **P**hase-conditioned **B**ehavior **D**ecoder in Sec. 3.3. Finally, the training strategy is explained in Sec 3.4.

## 3.1. Problem Formulation

Given expert demonstrations $\mathcal{D} = \{(\tau_i, L_i)\}_{i=1}^N$ with sequences of observations $O \in \mathcal{O}$ and actions $a \in \mathcal{A}$ conditioned on instruction $L$, standard VLAs directly regress $\pi(\mathbf{a}_{t:t+k}|O_t, L)$ in the ambient space, often lead to overfitting. Instead, we propose a latent variable model governed by hierarchical manifold coordinates: a **time-invariant global prototype** $z_{\text{proto}}$ (capturing task topology) and a **time-variant phase state** $z_{\text{phase}}$ (tracking execution progress). This factorization explicit decouples global structure from local dynamics:

$$p(\mathbf{a}_{t:t+k}|O_t, L) = \int \underbrace{p(\mathbf{a}_{t:t+k}|z_{\text{proto}}, z_{\text{phase}}, O_t, L)}_{\text{Manifold-Guided Execution}}$$
$$\cdot \underbrace{p(z_{\text{phase}}|O_t, a_{t-1})}_{\text{Phase Estimation}} \underbrace{p(z_{\text{proto}}|O_0, L)}_{\text{Prototype Retrieval}} \, dz \tag{1}$$

## 3.2. Visuomotor Behavior Encoder (VBE)

**Motivation.** Existing encoders often focus on frame-level features or short-horizon primitives, failing to capture the global causal dependencies essential for complex tasks. The VBE is designed to address this by functioning as a *selective information bottleneck*. Its goal is to filter high-frequency environmental noise while retaining the intrinsic low-frequency topological structure of the behavior.

### 3.2.1. CAUSAL THREE-STREAM ARCHITECTURE

To model trajectory dynamics, we introduce a hierarchical three-stream architecture comprising vision ($S_v$), action ($S_a$), and behavior ($S_z$) streams, which explicitly interleaves temporal causal modeling with spatial multimodal fusion.

**Global Temporal Modeling via Mamba.** To efficiently capture long-horizon dependencies, each stream ($m \in \{v, a, z\}$) adopts the Mamba (Gu & Dao, 2024) based on Selective State Space Model (SSM). We discretize the continuous latent dynamics via the Zero-Order Hold (ZOH) rule, governed by a timescale $\mathbf{\Delta}_t = \text{Softplus}(\text{Linear}(x_t^{(m)}))$. This yields time-varying discrete parameters:

$$\bar{\mathbf{A}}_t = \exp(\mathbf{\Delta}_t\mathbf{A}), \quad \bar{\mathbf{B}}_t = (\mathbf{\Delta}_t\mathbf{A})^{-1}(\bar{\mathbf{A}}_t - \mathbf{I})\mathbf{\Delta}_t\mathbf{B} \tag{2}$$

where $\mathbf{A}$ and $\mathbf{B}$ denote the continuous evolution and input parameters. Crucially, the *input-dependence* of $\mathbf{\Delta}_t$ empow-

ers the VBE to function as a *selective filter*, dynamically suppressing irrelevant observations (e.g., background clutter) while preserving critical task events in the latent state $h_t^{(m)}$. The state is recursively updated and projected via a gated connection:

$$h_t^{(m)} = \bar{\mathbf{A}}_t h_{t-1}^{(m)} + \bar{\mathbf{B}}_t \text{LayerNorm}(x_t^{(m)})$$
$$\tilde{h}_t^{(m)} = x_t^{(m)} + \text{Linear}(\mathbf{C}_t h_t^{(m)} \odot \sigma(g_t)) \tag{3}$$

where $\mathbf{C}_t$ represents the output projection parameter, $g_t$ denotes the gating branch, and $\sigma$ is the SiLU activation. This linear complexity $\mathcal{O}(L)$ is pivotal for scaling to long-horizon robotic demonstrations.

**Spatial Multimodal Fusion.** After independent temporal filtering, we employ a progressive interaction strategy to resolve semantic ambiguities. We align low-level semantics via mutual attention between the vision and action streams:

$$\tilde{h}_t^{(v)} \leftarrow \tilde{h}_t^{(v)} + \text{Attn}(Q = \tilde{h}_t^{(v)}, K = \tilde{h}_t^{(a)}, V = \tilde{h}_t^{(a)})$$
$$\tilde{h}_t^{(a)} \leftarrow \tilde{h}_t^{(a)} + \text{Attn}(Q = \tilde{h}_t^{(a)}, K = \tilde{h}_t^{(v)}, V = \tilde{h}_t^{(v)}) \tag{4}$$

Based on the interaction, the behavior stream queries the unified context to extract the global task structure:

$$\mathcal{K}_t = [\tilde{h}_t^{(v)}; \tilde{h}_t^{(a)}]$$
$$\tilde{h}_t^{(z)} \leftarrow \tilde{h}_t^{(z)} + \text{Attn}(Q = \tilde{h}_t^{(z)}, K = \mathcal{K}_t, V = \mathcal{K}_t) \tag{5}$$

By querying this joint distribution, the manifold stream functions as an information bottleneck, effectively filtering residual environmental noise while distilling the topological essence of the behavior.

### 3.2.2. MANIFOLD COORDINATE PARAMETERIZATION

The VBE decouples the encoded trajectory into two orthogonal coordinates for inference:

**Global Prototype** ($z_{\text{proto}}$). To capture scene-invariant task topology, we construct an offline memory bank $\mathbb{M}$ via temporal mean pooling of behavior tokens: $z_{\text{proto}} = \frac{1}{T} \sum_{t=1}^T \tilde{h}_t^{(z)}$. During inference, we retrieve the top-$K$ prototypes most relevant to the query $q = \text{MLP}(\Phi(O_0, L))$ and aggregate them via weighted pooling:

$$\hat{z}_{\text{proto}} = \sum_{i \in \mathcal{N}_K} \frac{\exp(\langle q, k_i \rangle / \kappa)}{\sum_{j \in \mathcal{N}_K} \exp(\langle q, k_j \rangle / \kappa)} \cdot z_{\text{proto}}^{(i)} \tag{6}$$

where $\mathcal{N}_K$ denotes the indices of the top-$K$ candidates. Encapsulating the core behavioral abstraction, $\hat{z}_{\text{proto}}$ functions as a global behavioral guide. By providing a stable skeleton for the PBD and a persistent context for the flow policy, it anchors the low-level execution to the high-level task intent, ensuring globally consistent planning.

**Local Phase** $z_{\text{phase}}$. To track *real-time* progress, the VBE recursively updates the phase state based on the current

observation $O_t$ and previous action $a_{t-1}$:

$$z_{\text{phase}}^{(t)} = \text{VBE}_{\text{causal}}(z_{\text{phase}}^{(t-1)}, O_t, a_{t-1}) \tag{7}$$

This ensures the model remains synchronized with physical execution, mitigating temporal misalignment.

### 3.3. Phase-Conditioned Behavior Decoder (PBD)

**Motivation.** Standard latent models typically decode actions from a static variable, lacking awareness of the agent's real-time status. This leads to temporal drift, where generated actions mismatch the current scene. To bridge this gap, PBD adopts a **Predictor-Corrector** paradigm: it first predicts a *phase-aligned structural prior* and then uses it to bias a *flow matching policy* for precise control.

#### 3.3.1. PHASE-GUIDED TOPOLOGY UNFOLDING

The Predictor aims to generate a coarse-grained action skeleton that is strictly aligned with the current execution phase. First, we unfold the global prototype $\hat{z}_{\text{proto}}$ into a sequence of latent anchors $\mathbf{M} \in \mathbb{R}^{H \times D}$ via a generator $\mathcal{G}_\phi$:

$$\mathbf{M} = \mathcal{G}_\phi(\hat{z}_{\text{proto}}) \oplus \mathbf{P}_{\text{pos}} \tag{8}$$

Here, $\mathbf{P}_{\text{pos}}$ induces a canonical temporal geometry, transforming semantic features into an ordered manifold skeleton. Subsequently, the phase state $z_{\text{phase}}^{(t)}$ serves as a continuous query to dynamically interpolate the local geometry $c_t$ from these discrete anchors:

$$c_t = \text{Progress-Attn}(Q = z_{\text{phase}}^{(t)}, K = \mathbf{M}, V = \mathbf{M}) \tag{9}$$

This mechanism performs a differentiable interpolation on the manifold, retrieving a context $c_t$ that reflects the local geometry of the task. Finally, $c_t$ is projected to parameterize a Gaussian action prior, providing a stable initialization:

$$p(\mathbf{a}_{t:t+k}|c_t) = \mathcal{N}(\mathbf{a}_{t:t+k}; \mu_\psi(c_t), \text{diag}(\exp(\sigma_\psi(c_t)))) \tag{10}$$

#### 3.3.2. GEOMETRY-GUIDED FLOW MATCHING

The Corrector employs a Conditional Flow Matching policy $\pi_\theta$ to refine the structural prior $\mu_{\text{prior}} \triangleq \mu_\psi(c_t)$ into precise control. To enforce topological consistency, we introduce a latent structural biasing mechanism. We inject the prior guidance directly into the noisy embedding space $e(a_\sigma)$ via the guidance strength $\lambda$:

$$\tilde{e}(a_\sigma) = e(a_\sigma) + \lambda \cdot \text{Proj}_\phi(\mu_{\text{prior}}) \tag{11}$$

The flow matching vector field $v_\theta$ is then conditioned on this biased representation to generate the trajectory differential with respect to the flow time $\sigma \in [0, 1]$:

$$da_\sigma = v_\theta(\tilde{e}(a_\sigma), \sigma, \Phi(O_t, L), \hat{z}_{\text{proto}})d\sigma, \quad a_1 \sim \mathcal{N}(0, \mathbf{I}) \tag{12}$$

Theoretically, this additive injection shifts the attention manifold toward high-probability regions defined by the prior, allowing the flow policy to focus on resolving local geometric details and dynamics.

### 3.4. Training Strategy

Our training framework proceeds in two phases, corresponding to the abstraction and instantiation capabilities required for robust VLA models.

#### 3.4.1. PHASE1: BEHAVIOR MANIFOLD LEARNING

We train the VBE to compress high-dimensional sensorimotor sequences $\tau = \{(O_t, a_t)\}_{t=1}^T$ into compact behavior tokens using a composite objective:

$$\mathcal{L}_{\text{Stage1}} = \mathcal{L}_{\text{rec}} + \alpha \mathcal{L}_{\text{global}} + \beta \mathcal{L}_{\text{local}} \tag{13}$$

**Joint Predictive Reconstruction.** To distill essential task dynamics from visual redundancy, we adopt the Joint Embedding Predictive Architecture (JEPA). Let $\Phi$ be the vision encoder and $\Phi_{\text{ema}}$ be its exponentially moving average target. We minimize the joint prediction error of the future latent state and action:

$$\mathcal{L}_{\text{rec}} = \sum_t \underbrace{\|\hat{a}_t - a_{t+1}\|^2}_{\text{Action Prediction}} + \underbrace{\|\hat{v}_t - \text{SG}(\Phi_{\text{ema}}(O_{t+1}))\|^2}_{\text{Latent State Prediction}} \tag{14}$$

where $\text{SG}[\cdot]$ denotes the stop-gradient operator. This dual objective enforces complementary constraints: the visual term distills transition dynamics from static redundancy, while the action term anchors the representation in the control space. This ensures the learned manifold is both physically consistent and behaviorally actionable.

**Global Task Clustering.** We employ a supervised contrastive loss to organize the manifold semantically. For a trajectory $i$ in batch $\mathcal{B}$, we maximize its similarity with positive peers $\mathcal{P}(i)$ sharing the same behavior label, clustering functionally similar behaviors:

$$\mathcal{L}_{\text{global}} = \sum_{i \in \mathcal{B}} \frac{-1}{|\mathcal{P}(i)|} \sum_{p \in \mathcal{P}(i)} \log \frac{\exp(z_{\text{proto}}^{(i)} \cdot z_{\text{proto}}^{(p)}/\gamma)}{\sum_{k \in \mathcal{B} \setminus \{i\}} \exp(z_{\text{proto}}^{(i)} \cdot z_{\text{proto}}^{(k)}/\gamma)} \tag{15}$$

**Local Progress Distinctiveness.** To ensure latent tokens encode precise execution progress, we enforce temporal distinctiveness via InfoNCE. By treating distinct timesteps $t' \neq t$ as negative samples, we prevent topological collapse:

$$\mathcal{L}_{\text{local}} = -\sum_t \log \frac{\exp(z_t \cdot z_t/\tau)}{\sum_{t'} \exp(z_t \cdot z_{t'}/\tau)} \tag{16}$$

#### 3.4.2. PHASE 2: PRIOR-GUIDED POLICY TUNING

In the second stage, we jointly optimize the flow policy and the PBD using a composite objective:

$$\mathcal{L}_{\text{Stage2}} = \mathcal{L}_{\text{flow}} + \lambda_{\text{prior}} \mathcal{L}_{\text{prior}} \tag{17}$$

**Conditional Flow Matching.** To train the vector field $v_\theta$ defined in Sec. 3.3.2, we construct an Optimal Transport

*Table 1.* Performance comparison on RoboTwin 2.0 (Chen et al., 2025c). We report per-task success rates (SR) over 100 rollouts under *Hard* setting. [†] represents the result we reproduced. 20 tasks results are available in **Appendix**.

| Method | RoboTwin 2.0 (Hard) | | | | | | | | | |
|---|---|---|---|---|---|---|---|---|---|---|
| | Adjust Bottle | Click Alarmclock | Click Bell | Dump Bin Bigbin | Grab Roller | Move Playingcard | Place a2b Right | Place Bread Basket | Place Burger Fries | Place Container Plate |
| **RDT** (Liu et al., 2024c) | 75% | 12% | 9% | 32% | 43% | 11% | 1% | 2% | 27% | 17% |
| **ACT** (Zhao et al., 2023a) | 23% | 4% | 3% | 1% | 25% | 0% | 0% | 0% | 0% | 1% |
| **DP** (Chi et al., 2025b) | 0% | 5% | 0% | 0% | 0% | 0% | 0% | 0% | 0% | 0% |
| **DP3** (Ze et al., 2024) | 3% | 14% | 0% | 53% | 2% | 3% | 0% | 1% | 18% | 1% |
| $\pi_0$ (Black et al., 2024) | 56% | 11% | 3% | 24% | 80% | 22% | 6% | 4% | 4% | 45% |
| $\pi_{0.5}$[†] (Intelligence et al., 2025) | 75% | 44% | 64% | 69% | 82% | 32% | 19% | 28% | 46% | 55% |
| **BehaviorVLA** | **83%** | **52%** | **77%** | **77%** | **90%** | **41%** | **25%** | **36%** | **61%** | **62%** |

*Table 2.* Results on the LIBERO benchmark (Liu et al., 2023). We report Success Rate across the Spatial, Object, Goal, and Long suites, along with the average performance. Best results are highlighted in **bold**.

| Method | Spatial | Object | Goal | Long | Avg. |
|---|---|---|---|---|---|
| Diffusion Policy (Chi et al., 2025b) | 78.5 | 87.5 | 73.5 | 64.8 | 76.1 |
| gr00t-N1 (Bjorck et al., 2025) | 94.4 | 97.6 | 93.0 | 90.6 | 93.9 |
| OpenVLA (Kim et al., 2024) | 84.7 | 88.4 | 79.2 | 53.7 | 76.5 |
| OpenVLA-OFT (Kim et al., 2025) | 97.6 | 98.4 | 97.9 | 94.5 | 97.1 |
| MemoryVLA (Shi et al., 2025) | 98.4 | 98.4 | 96.4 | 93.4 | 96.7 |
| UniVLA (Bu et al., 2025) | 95.4 | 98.8 | 93.6 | 94.0 | 95.4 |
| InternVLA-M1 (Chen et al., 2025d) | 98.0 | 99.0 | 93.8 | 92.6 | 95.9 |
| $\pi_0$-Fast (Pertsch et al., 2025) | 96.4 | 96.8 | 88.6 | 60.2 | 85.5 |
| $\pi_0$ (Black et al., 2024) | 96.8 | 98.8 | 95.8 | 85.2 | 94.2 |
| $\pi_{0.5}$ (Intelligence et al., 2025) | 98.8 | 98.2 | 98.0 | 92.4 | 96.9 |
| **BehaviorVLA** | **99.2** | **99.4** | **98.8** | **94.6** | **98.0** |

*Table 3.* Ablation on LIBERO and Real-World. We report Success Rate on LIBERO and Real-World.

| Method | | LIBERO | | | | Real-World | |
|---|---|---|---|---|---|---|---|
| VBE | PBD | Spatial | Object | Goal | Long | Gen. | Long. |
| | | 98.8 | 98.2 | 98.0 | 92.4 | 57.0 | 41.0 |
| ✓ | | 98.2 | 98.8 | **98.8** | 93.8 | 65.0 | 48.0 |
| | ✓ | 99.0 | 99.0 | 98.4 | 93.4 | 60.0 | 45.0 |
| ✓ | ✓ | **99.2** | **99.4** | **98.8** | **94.6** | **70.0** | **55.0** |

conditional path. Let $\mathbf{a}_0$ be the ground-truth action from the dataset and $\mathbf{a}_1 \sim \mathcal{N}(\mathbf{0}, \mathbf{I})$ be the source noise. The training trajectory $\mathbf{a}_\sigma$ and its target velocity $u_\sigma$ are defined as:

$$\mathbf{a}_\sigma = \sigma \mathbf{a}_1 + (1 - \sigma)\mathbf{a}_0, \quad u_\sigma = \mathbf{a}_1 - \mathbf{a}_0 \qquad (18)$$

During training, we replace the deterministic guidance strength $\lambda$ with a stochastic dropout mask $m \sim \text{Bernoulli}(p)$ to prevent posterior collapse. The flow matching loss is computed on the geometry-guided embedding:

$$\mathbf{h}_\sigma = e(\mathbf{a}_\sigma) + m \cdot \text{Proj}_\phi(\mu_{\text{prior}})$$
$$\mathcal{L}_{\text{flow}} = \mathbb{E}_{\sigma, \mathbf{a}_1, \mathbf{a}_0, m}\left[\|v_\theta(\mathbf{h}_\sigma, \sigma, \Phi(O_t, L), \hat{z}_{\text{proto}}) - (\mathbf{a}_1 - \mathbf{a}_0)\|^2\right]$$
$$(19)$$

Note that the regression target drives the flow from data $\mathbf{a}_0$ to noise $\mathbf{a}_1$, aligning with the inference process where we integrate from $\sigma = 1$ to $\sigma = 0$.

**Manifold-Constrained Prior Learning.** The PBD provides the structural initialization $\mu_{\text{prior}}$. We supervise it by minimizing the Negative Log-Likelihood (NLL) of the expert actions $\mathbf{a}_0$ under the predicted distribution:

$$\mathcal{L}_{\text{prior}} = -\mathbb{E}\left[\sum_{k=1}^{H} \log \mathcal{N}(\mathbf{a}_0^{(k)} \mid \mu_{\text{prior}}^{(k)}, \sigma_{\text{prior}}^{(k)})\right] \qquad (20)$$

## 4. Experiment

We conduct extensive experiments across both simulation benchmarks and real-world scenarios to comprehensively evaluate BehaviorVLA. Our evaluation is designed to assess not only core manipulation proficiency but also the model's robustness and generalization capabilities under diverse and challenging conditions.

### 4.1. Evaluation on Simulation Benchmarks

**Simulation Benchmarks.** To assess robustness of BehaviorVLA under varied simulated conditions, we conduct experiments on RoboTwin2.0 (Chen et al., 2025b), LIBERO (Liu et al., 2023), and CALVIN (Mees et al., 2022). For RoboTwin 2.0, we evaluate on 20 randomly selected tasks and 100 rollouts in the Hard setting (domain randomization in clutter, lighting, textures, and height). For LIBERO (Liu et al., 2023), we report the average success rate across the Spatial, Object, Goal, and Long suites (each comprising 10 tasks), evaluated over 500 rollouts per suite. For CALVIN, we follow standard $ABC \rightarrow D$ setting to validate the model's generalization capabilities across scenes and objects. Detailed CALVIN experimental results and analysis are provided in the **Appendix**.

**Implementation Details.** BehaviorVLA is built upon the $\pi_{0.5}$ backbone (Intelligence et al., 2025), integrating the proposed Visuomotor Behavior Encoder (VBE) and Phase-conditioned Behavior Decoder (PBD). We fine-tune the model for 30k steps on a cluster of 8× A800 GPUs with a batch size of 256. The learning rate is set to $5 \times 10^{-5}$. Com-

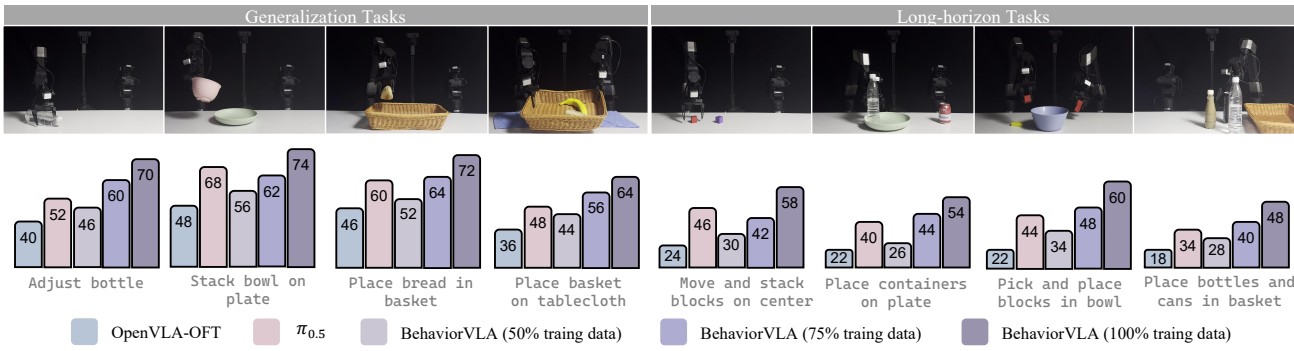

Figure 3. Real-world task setup and evaluation results. BehaviorVLA outperforms OpenVLA-OFT(Kim et al., 2025) and $\pi_{0.5}$ (Intelligence et al., 2025) across both generalization and long-horizon tasks. Notably, BehaviorVLA demonstrates superior data efficiency, maintaining competitive performance even when trained with reduced dataset sizes (50% and 75%).

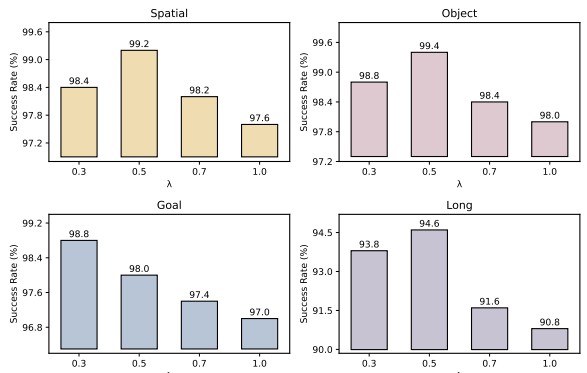

Figure 4. Ablation on Guidance Strength $\lambda$ in the inference. An optimal guidance strength is essential. Either insufficient or excessive $\lambda$ leads to degradation.

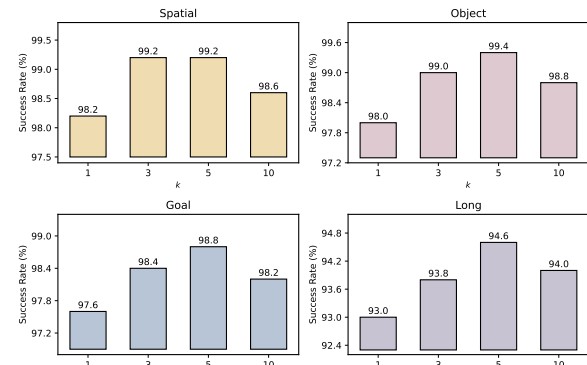

Figure 5. Ablation on the number of retrieved prototypes $k$. The results show that $k = 5$ provides a favorable balance between prototype diversity and retrieval precision across the four suites.

prehensive hyperparameter settings and details are provided in the **Appendix**.

**Results on RoboTwin 2.0**. On the RoboTwin 2.0 benchmark (Table 1), BehaviorVLA achieves average success rate of 58% in the *Hard* setting, outperforming RDT (Liu et al., 2024c) by +37.7%. Complex bimanual arms tasks impose high demands on the VLA model's motion accuracy and coordination capabilities. BehaviorVLA solves this via General-to-Specific Instantiation. The PBD operates as a Predictor-Corrector: it first unfolds a stable behavior skeleton and then refines it via Flow Matching. This mechanism ensures that the instantiated actions are not only globally coordinated but also locally precise, effectively handling complex dynamics.

**Results on LIBERO**. As shown in Table 2, BehaviorVLA achieves an average success rate of 98%, outperforming existing state-of-the-art methods across all task suites, including $\pi_{0.5}$ (Intelligence et al., 2025) and OpenVLA-OFT (Kim et al., 2025). Notably, the most substantial improvement is observed LIBERO-Long (+2.2%). We attribute this

robustness to the synergy of our proposed modules: the VBE constructs a topologically organized manifold that prevents temporal drift over extended sequences, while the PBD injects phase-aware structural guidance into the flow policy. This ensures that the model maintains global task coherence without sacrificing the local geometric precision.

### 4.2. Evaluation on Real-World

To validate the efficacy of BehaviorVLA in physical environments, we conduct experiments on the GALAXEA R1 Lite, a 14-DoF bimanual platform equipped with both wrist-mounted and third-person cameras ($224 \times 224$ resolution). Our evaluation protocol encompasses two distinct categories: *Generalization Tasks* and *Long-horizon Tasks*. For the former, we collect 100 expert demonstrations per task and perform 50 evaluation rollouts, with varying lighting conditions and scene configurations. For the latter, we utilize 150-200 demonstrations and conduct 50 rollouts per task. In all evaluations, object positions are randomly initialized to prevent overfitting. More details are provided in the **Appendix**.

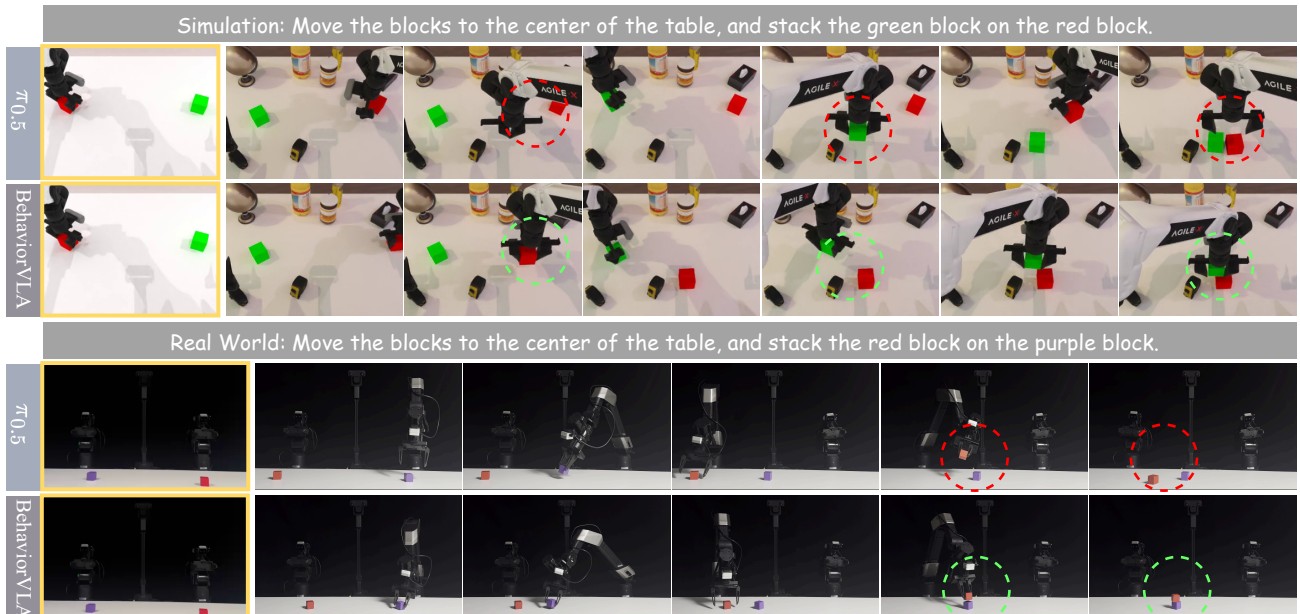

*Figure 6.* Qualitative comparison of simulation and real-world manipulation tasks. Yellow bounding boxes indicate the training scenarios. **Top**: In simulation, the baseline $\pi_{0.5}$ (Black et al., 2025) fails to grasp the target block when subjected to variations in background and object position. **Bottom**: In real-world experiments, our **BehaviorVLA** demonstrates strong few-shot transfer capabilities, accurately completing the task despite variations in block size, color, and position.

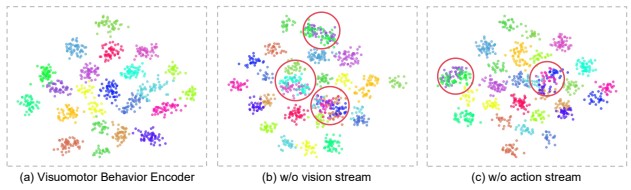

*Figure 7.* **t-SNE Visualization.** (a) The VBE shows clear, distinct behavior clusters. Removing (b) the vision stream or (c) the action stream causes clusters to mix and scatter. This highlights our tri-stream design is essential for learning highly discriminative behavior representations.

**Results on Real-World**. As illustrated in Figure 3, BehaviorVLA demonstrates superior performance, achieving average success rates of 70% and 55% on *Generalization* and *Long-horizon* tasks, respectively. In the *Generalization Tasks*, Behavior exhibits remarkable robustness to variations in lighting, scenes, and positions. We attribute this to the Specific-to-General Abstraction mechanism: the VBE effectively filtering out high-frequency environmental noise while preserving the scene-invariant topological structure of the task. For *Long-horizon Tasks*, BehaviorVLA outperforms the strong baseline $\pi_{0.5}$ by a significant margin of 34%. This gain highlights the critical role of the Phase-conditioned Behavior Decoder (PBD). By continuously aligning the action generation with the real-time execution phase, PBD prevents the temporal drift often observed in standard policies, ensuring consistent and globally coherent bimanual control.

## 4.3. Ablation Study

**Ablation on VBE and PBD.** As shown in Table 3, removing the VBE causes a performance drop of 16% on Real-World. Without VBE's abstraction, the model overfits to environmental noise rather than capturing the invariant task structure, significantly impairing generalization. Similarly, removing the PBD leads to a 9.6% decline on Real-World. Lacking the phase-aligned prior, the model fails to track real-time execution progress, causing generated actions to drift and fail during long-horizon tasks.

**Ablation on Guidance Strength.** We further investigate the impact of the prior guidance strength $\lambda$. As shown in Figure 4, the success rate reaches its peak at an optimal $\lambda$, while deviating from this value in either direction results in performance degradation. When $\lambda$ is too small, the policy lacks sufficient structural guidance, leading to inconsistent trajectory generation. Conversely, an excessively large $\lambda$ imposes an over-constraining prior that suppresses the fine-grained local corrections necessary for precise interaction. This confirms that a properly tuned $\lambda$ is essential to balance global structural stability with local control flexibility.

**Ablation on the Number of Retrieved Prototypes.** We study the effect of the retrieval number $k$ in the behavior memory. As shown in Figure 5, the performance reaches its peak at $k = 5$, while either smaller or larger values lead to degradation. A small $k$ provides insufficient behavioral diversity and makes the retrieved prior sensitive to individual

query bias. In contrast, an overly large $k$ introduces less relevant prototypes, which may disturb the global structural guidance and degrade action generation. This confirms that an appropriate $k$ is important for balancing retrieval diversity and prototype relevance.

### 4.4. Qualitative Analysis

In Figuer 6, we present qualitative examples from both simulation and real-world scenarios. The top panel shows the RoboTwin task under varying backgrounds and object positions. $\pi_{0.5}$ tends to overfit to the training environments, leading to failures when encountering variations in spatial layout. In contrast, BehaviorVLA leverages VBE to filter irrelevant clutter, focusing on essential object interactions to ensure robust success despite perturbations. The bottom panel illustrates sim-to-real adaptation with few-shot learning (50 demostration). $\pi_{0.5}$ fails to rapidly adapt to variations in scene layout, object attributes, and lighting conditions. BehaviorVLA rapidly adapts to novel scenarios via behavior representation, ensuring smooth and precise manipulation. More examples are provided in **Appendix**.

### 4.5. Visualization of Behavioral Representation

To evaluate the quality of the learned behavior representations, we visualize the global prototypes of 20 distinct task modes using t-SNE. The results clearly demonstrate the superiority of our tri-stream architecture in producing discriminative representations. Specifically, removing the vision stream ($S_v$) leads to semantic ambiguity (Figure 7b), where behaviors with identical motion but different visual semantic (e.g., *Stirring a pot* vs. *Wiping a table*) collapse into overlapping clusters. Conversely, excluding the action stream ($S_a$) prevents the model from capturing the temporal action patterns of the behavior (Figure 7c). Without the causal history of actions, the representation degenerates into a static visual descriptor, making it impossible to distinguish between tasks that appear visually similar but require different manipulation dynamics. This confirms that both visual context and action sequences are indispensable for learning robust and well-separated behavior representations.

## 5. Conclusion

In this work, we introduce BehaviorVLA, a framework designed to enhance the robustness of VLA models by learning temporally coherent behavior representations. Through the synergistic design of the Visuomotor Behavior Encoder and Phase-conditioned Behavior Decoder, our approach effectively balances global behavior abstraction with precise, phase-aligned control. Extensive experiments demonstrate that BehaviorVLA achieves state-of-the-art performance across simulation benchmarks and significantly enhances sim-to-real transfer efficiency, matching leading baselines

with only 50% of the fine-tuning data. These results suggest that explicitly modeling structured behavior representations is a scalable and data-efficient path toward robust robotic manipulation in Real-World.

## Limitation and Future Work

Although BehaviorVLA demonstrates superior robustness and data efficiency in sim-to-real transfer through the Visuomotor Behavior Encoder and Phase-conditioned Behavior Decoder, several limitations remain. First, the effectiveness of the specific-to-general abstraction is inherently constrained by the topological diversity of the offline prototype memory. When a novel task substantially departs from the learned behavior manifold, the retrieved structural skeleton may be ill-posed, potentially guiding the PBD toward geometrically consistent but functionally incorrect actions. Second, the general-to-specific instantiation relies on iterative differential equation solving for flow matching. While the Predictor-Corrector paradigm improves precision, the numerical integration incurs higher inference latency compared to simple regression baselines, which may challenge high-frequency control on compute-constrained hardware.

A natural direction is to develop online manifold expansion mechanisms, where the prototype memory bank can be dynamically updated through self-supervised exploration and interaction feedback, enabling the model to adapt to out-of-distribution task topologies. Another avenue is to apply consistency distillation to the Phase-conditioned Behavior Decoder, compressing the iterative flow generation into a single-step inference while preserving the topological alignment provided by the Visuomotor Behavior Encoder. These remain as future work for this paper.

## Impact Statement

This paper presents work whose goal is to advance the field of Machine Learning. There are many potential societal consequences of our work, none which we feel must be specifically highlighted here.

## Acknowledgements

This study is supported by National Natural Science Foundation of China (Grant No. 62306090) , Natural Science Foundation of Guangdong Province of China (Grant No. 2024A1515010147) and Natural Science Foundation of Shenzhen City of China (Grant No. JCYJ20250604145700001) and Beijing Natural Science Foundation (L254018).

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

# A. Related Work

**Vision-Language-Action Models.** Building upon the advancements of Vision-Language Models (VLMs) (Alayrac et al., 2022; Karamcheti et al., 2024), Vision-Language-Action (VLA) models(Kim et al., 2024; Zitkovich et al., 2023; Black et al., 2025; Li et al., 2025c; Shao et al., 2025; Li et al., 2026) have emerged as a promising paradigm in robot learning. Recent works have further extended VLA capabilities through the integration of enhanced visual perception(Li et al., 2025a; Qu et al., 2025; Liu et al., 2025b), efficient paradigms (Liu et al., 2024b; Chen et al., 2025a), and dual-system architectures (Bjorck et al., 2025; Wang et al., 2025; Wen et al., 2025). However, existing VLAs rely on implicit encodings that fail to capture the structural isomorphism intrinsic to manipulation, rendering models fragile to geometric shifts and contact dynamics. To address this, we introduce the Visuomotor Behavior Encoder (VBE), which constructs a structured prototype memory to explicitly model behavior geometry, ensuring robust and data-efficient adaptation to geometric variations.

**Memory-Retrieval Mechanisms for Robot Learning.** To handle partial observability and generalization in manipulation, recent robot learning models increasingly use retrieval-based memory. MemoryVLA(Shi et al., 2025) equips policies with working and long-term memory to support manipulation, while EchoVLA(Lin et al., 2026) represents scene and episodic information as declarative memory for retrieval and fusion. MAP-VLA(Li et al., 2025b) further reduces fragment inconsistency through stage-wise segmentation and alignment. Related ideas also appear in embodied agents and generalist policies (Zhu et al., 2024; Anwar et al., 2025; Xie et al., 2024), which retrieve trajectories or policies from external memory to guide behavior in new environments. However, existing methods lack explicit progress modeling and treat retrieved trajectories as static context, leading to phase misalignment and action averaging. VBE addresses this limitation by introducing prototype latents for behavior retrieval and phase latents for progress tracking, enabling phase-consistent and progress-aware VLA.

**Generative Policies for Robot Control.** To mitigate mode averaging in imitation learning (An et al., 2025), generative policies like diffusion policy (Chi et al., 2025a) and flow policy (Rouxel et al., 2024) have become the standard for robot control, modeling generation as a transport process from gaussian noise to multi-modal distributions. Recent VLAs(Black et al., 2025; Black et al.; Shukor et al., 2025) extend this via hierarchical VLM-action expert architectures. However, relying on uninformative priors poses fundamental limitations in contact-rich tasks: (1) lack of progress awareness: stochastic sampling without phase constraints causes latent stage jumping and temporal inconsistency. (2) contact instability: iterative generation noise induces high-frequency jitter and premature actuation, destabilizing physical interaction. To address this, we introduce a Phase-Conditioned Prior Decoder that generates a phase-consistent prior as the predictor for structural guidance, coupled with a flow policy as the corrector for residual dynamics.

# B. Implementation and Training Details

In this section, we provide detailed specifications regarding the data preprocessing, model architecture, and optimization strategy. Our training pipeline consists of two distinct stages: (1) learning the behavior manifold via the Visuomotor Behavior Encoder (VBE), and (2) prior-guided policy tuning of the $\pi_{0.5}$ backbone with Phase-Conditioned Behavior Decoder (PBD). We utilize the RoboTwin2.0 benchmark as the representative setup. Specifically, we employ the *clean* subset of expert demonstrations from the 50 standard tasks in RoboTwin 2.0 as our training data.

## B.1. Phase 1: Behavior Manifold Learning

B.1.1. DATA PREPROCESSING AND ORGANIZATION.

We resize all visual observations to a resolution of $224 \times 224$. The action space for RoboTwin2.0 consists of dual-arm end-effector poses and gripper states, resulting in a dimensionality of $D_a = 14$.

**Task-Specific Normalization:** A critical component of our pipeline is task-specific normalization. For each distinct sub-task (identified by its directory structure), we compute independent statistics (mean $\mu$ and standard deviation $\sigma$) for actions and proprioceptive states. During training, samples are normalized using the statistics corresponding to their specific task ID, ensuring that manifold learning is invariant to the magnitude of action spaces across disparate tasks.

**Sequence Handling:** We employ a frame skip of $k = 4$ to reduce temporal redundancy. To accommodate variable-length trajectories within a training batch, we apply zero-padding to align sequences to the maximum length in the batch. Corresponding boolean masks are generated to exclude padded regions from loss computation.

**Behavior Class Definition:** To establish the global consistency objective, we align our behavior classes with the 50 distinct

manipulation tasks defined in the RoboTwin 2.0 benchmark (e.g., *Stack Block*, *Adjust Bottle*). These semantic categories correspond directly to the 50 sub-directories within the dataset root. Trajectories from the same task category are treated as positive pairs regardless of object or configuration variations, while trajectories from different categories serve as negative samples. This compels the encoder to abstract the underlying task topology while discarding task-irrelevant environmental variations.

### B.1.2. MODEL ARCHITECTURE.

The Visuomotor Behavior Encoder follows a causal three-stream architecture with the following specifications:

**Backbone Configuration:** The model consists of $L = 8$ causal three-stream blocks. Within each block, the Vision, Action, and Behavior streams are processed by independent Mamba layers. The Selective State Space Model (SSM) parameters are set to: state dimension $d_{state} = 16$, convolutional width $d_{conv} = 4$, and expansion factor $E = 2$. The hidden model dimension is set to $D = 256$.

**Vision Encoders:** We utilize a LightweightCNN to extract visual features, projected to $D = 256$. A Target Vision Encoder is maintained via Exponential Moving Average (EMA) with a decay rate of $\tau = 0.99$ to facilitate the JEPA-style predictive objective.

**Projection Heads:** Both the Task Identity Head and Temporal Progress Head are implemented as two-layer MLPs with a hidden dimension of 256 and an output dimension of 128 for InfoNCE computation.

### B.1.3. OPTIMIZATION STRATEGY.

The VBE is implemented in PyTorch and trained on A800 (80G) GPUs using the AdamW optimizer with a weight decay of $1 \times 10^{-4}$. We employ a *Layer-wise Learning Rate* strategy: the vision encoder learning rate is set to $1 \times 10^{-5}$ to preserve pretrained features, while the causal three-stream architecture uses $1 \times 10^{-4}$ for rapid adaptation. Gradient clipping is applied with a maximum norm of 1.0. The model is trained with a batch size of 16 for 40 epochs.

### B.2. Phase 2: Prior-Guided Policy Tuning

We perform joint tuning of the pre-trained PaliGemma-based $\pi_{0.5}$ backbone and the Phase-Conditioned Behavior Decoder (PBD). To maximize computational efficiency, the hierarchical behavior coordinates—global prototype $z_{\text{proto}}$ and local phase state $z_{\text{phase}}$—are **pre-extracted offline** using the frozen VBE (trained in Phase 1) and cached in a memory bank. During training, these features are indexed via episode and frame identifiers, avoiding redundant on-the-fly computation.

### B.2.1. TRAINING FRAMEWORK

The training framework integrates two synergistic mechanisms:

**Action Prior Estimation:** The PBD serves as a coarse-level planner. It processes the retrieved hierarchical latents to predict a Gaussian distribution over the future action trajectory. This module is supervised via the Negative Log-Likelihood (NLL) of the ground truth actions, providing a structural initialization for the policy.

**Residual Injection with Stochastic Dropout:** To guide fine-grained generation, the mean action prediction from the PBD is projected and added as a *residual embedding* to the noisy action input of the $\pi_{0.5}$ Flow Matching backbone. Crucially, to prevent the policy from over-relying on this prior shortcut and ignoring current visual observations (i.e., posterior collapse), we apply **stochastic dropout** (Bernoulli masking) to the prior embedding during training. The entire architecture is optimized end-to-end by minimizing a joint objective composed of the conditional flow matching loss and the prior NLL loss.

### B.2.2. OPTIMIZATION SETUP.

The model is trained on 8 NVIDIA A800 GPUs with a global batch size of 256. We utilize the AdamW optimizer with a constant learning rate of $5 \times 10^{-5}$. The training process spans 30,000 steps to ensure the convergence of both the fine-grained flow matching objective and the coarse-level prior distribution.

## C. Evaluation

### C.1. Evaluation on CALVIN

CALVIN (Mees et al., 2022) is an open-source, PyBullet-based simulated benchmark for long-horizon language-conditioned robotic manipulation, instantiated across four visually diverse yet structurally aligned tabletop environments (A–D) that feature a 7-DoF Franka Emika Panda arm with a parallel gripper operating in a workspace containing a sliding door, a drawer, a push-button LED, a toggle switch for a light bulb, and three colored blocks. Within this setup, CALVIN defines 34 automatically detectable manipulation tasks, and provides multimodal onboard observations—including RGB-D inputs from both a static and an in-hand camera, proprioceptive state, and optional vision-based tactile sensing—together with continuous control at 30 Hz. We follow the Long-Horizon MTLC setting: 1000 filtered instruction chains of length 5 with neutral pose resets, thereby probing generalization to novel linguistic paraphrases and unseen environments while requiring robust multi-skill composition. The evaluation metrics include average sequence length and success rates over 500 rollouts.

As shown in Table 4, BehaviorVLA attains an average length of 4.36 on CALVIN, surpassing $\pi_0$ (Black et al., 2024) by 11%. In the challenging ABC $\to$ D setting (unseen environments), standard VLA models suffer from the "distribution shift" problem, overfitting to specific simulation textures. In contrast, our results validate the effectiveness of the Specific-to-General Abstraction. By distilling trajectories into a scene-invariant behavior manifold, VBE filters out environmental noise (e.g., background clutter), allowing the model to transfer the core task topology to novel scenes robustly.

*Table 4.* Performance comparison on CALVIN (Mees et al., 2022). We report the Success Rate of each track and average completion length (Avg. Len). [†] represents the result we reproduced.

| Method | CALVIN (ABC $\to$ D) | | | | | |
|---|---|---|---|---|---|---|
| | 1/5 | 2/5 | 3/5 | 4/5 | 5/5 | Avg. Len |
| RoboVLM (Liu et al., 2025a) | **98.0** | **93.6** | 85.4 | 77.8 | 70.4 | 4.25 |
| ReconVLA (Song et al., 2025) | 95.6 | 87.6 | 76.9 | 69.3 | 64.1 | 3.95 |
| OpenVLA (Kim et al., 2024) | 91.3 | 77.8 | 62.0 | 52.1 | 43.5 | 3.27 |
| UniVLA (Bu et al., 2025) | 95.5 | 85.8 | 75.4 | 66.9 | 56.5 | 3.80 |
| UP-VLA (Zhang et al., 2025) | 92.8 | 86.5 | 81.5 | 76.9 | 69.9 | 4.08 |
| RoboDual (Bu et al., 2024) | 94.4 | 82.7 | 72.1 | 62.4 | 54.4 | 3.66 |
| Seer (Tian et al., 2024) | 96.3 | 91.6 | 86.1 | 80.3 | 74.0 | 4.28 |
| VPP (Hu et al., 2024) | 95.7 | 91.2 | 86.3 | 81.0 | 75.0 | 4.29 |
| $\pi_0$ (Black et al., 2024) | 93.8 | 85.0 | 76.7 | 68.1 | 59.9 | 3.92 |
| $\pi_{0.5}$[†] (Intelligence et al., 2025) | 94.8 | 88.9 | 84.1 | 79.7 | 72.1 | 4.21 |
| **BehaviorVLA** | 96.0 | 92.0 | **87.3** | **82.9** | **77.3** | **4.36** |

### C.2. Evaluation on RoboTwin 2.0

RoboTwin 2.0 presents a standardized bimanual manipulation benchmark leveraging the RoboTwin-OD object library, which comprises 731 annotated objects across 147 categories and a large-scale dataset of over 100k expert dual-arm trajectories. The benchmark encompasses 50 collaborative dual-arm tasks instantiated on five distinct robot embodiments. In the standard simulation protocol, each task is trained in a single-task manner on the Aloha–AgileX dual-arm platform using 50 clean expert demonstrations, while evaluation typically involves 100 rollouts under two difficulty tiers: an *Easy* regime with clutter-free scenes and a *Hard* regime characterized by strong domain randomization (including clutter, background textures, lighting, and tabletop height).

In this work, we evaluate a random subset of 20 tasks. Specifically, we train our models using all available clean demonstrations and conduct 100 evaluation rollouts per task under the *Hard* setting to rigorously test robustness against domain shifts. As reported in Table 5, BehaviorVLA achieves an average success rate of 58% across these 20 tasks, surpassing all baselines, including $\pi_{0.5}$.

### C.3. Evaluation on Real-World

We conduct real-world evaluations using the Galaxea R1 Lite, a bimanual mobile humanoid platform tailored for human-centric indoor environments. The robot possesses a 23-DoF kinematic configuration, comprising two 6-DoF arms equipped

*Table 5.* Performance comparison on RoboTwin 2.0. We report per-task success rates (SR) over 100 rollouts under *Hard* setting. [†] represents the result we reproduced.

| Task | RDT | ACT | DP | DP3 | $\pi_0$ | $\pi_{0.5}$[†] | BehaviorVLA |
|------|-----|-----|-----|-----|-----|-----|-----|
| Adjust Bottle | 75% | 23% | 0% | 3% | 56% | 75% | **83%** |
| Click Alarmclock | 12% | 4% | 5% | 14% | 11% | 44% | **52%** |
| Click Bell | 9% | 3% | 0% | 0% | 3% | 64% | **77%** |
| Dump Bin Bigbin | 32% | 1% | 0% | 53% | 24% | 69% | **77%** |
| Grab Roller | 43% | 25% | 0% | 2% | 80% | 82% | **90%** |
| Move Playingcard Away | 11% | 0% | 0% | 3% | 22% | 32% | **41%** |
| Place a2b Left | 1% | 0% | 0% | 2% | 1% | 20% | **30%** |
| Place a2b Right | 1% | 0% | 0% | 0% | 6% | 19% | **25%** |
| Place Bread Basket | 2% | 0% | 0% | 1% | 4% | 28% | **36%** |
| Place Burger Fries | 27% | 0% | 0% | 18% | 4% | 46% | **61%** |
| Place Container Plate | 17% | 1% | 0% | 1% | 45% | 55% | **62%** |
| Place Empty Cup | 7% | 0% | 0% | 1% | 11% | 59% | **67%** |
| Place Object Stand | 5% | 0% | 0% | 0% | 11% | 46% | **54%** |
| Place Shoe | 7% | 0% | 0% | 2% | 6% | 20% | **28%** |
| Place Stapler | 24% | 6% | 0% | 3% | 29% | 55% | **62%** |
| Rotate QRCode | 5% | 0% | 0% | 1% | 15% | 20% | **29%** |
| Shake Bottle Horizon | 51% | 4% | 18% | 25% | 51% | 85% | **92%** |
| Shake Bottle | 45% | 10% | 8% | 19% | 60% | 82% | **93%** |
| Stack Blocks Two | 2% | 0% | 0% | 0% | 1% | 21% | **32%** |
| Stack Bowls Two | 30% | 0% | 0% | 6% | 41% | 62% | **69%** |

with spherical wrists and parallel jaw grippers, a 3-DoF torso facilitating vertical and pitch adjustments for workspace extension, and a 6-DoF vector-drive omnidirectional base for coordinated whole-body manipulation.

To comprehensively assess model performance, we establish two distinct task suites, as detailed in Table 6 and Table 7: *Generalization Tasks* and *Long-horizon Tasks*. The former evaluates the model's robustness against variations in scene layout, lighting conditions, and object instances, while the latter focuses on temporal stability and consistency during extended execution sequences. For all evaluations, object poses are randomly initialized to prevent memorization. Quantitative results presented in Table 6 and Table 7 demonstrate that BehaviorVLA achieves superior performance across both suites. Qualitative visualizations are provided in the subsequent section.

*Table 6.* Overview of *Generalization Tasks* on Real-World.

| | **Generalization Tasks** | | | |
|---|---|---|---|---|
| **Task name** | adjust bottle | stack bowl on plate | place bread in basket | place basket on tablecloth |
| **Description** | Identify the orientation of a fallen bottle, grasp it, and reorient it to a vertical standing position. | Grasp a bowl from the table and stably stack it on top of a specific plate. | Pick the bread on the table and gently place it inside a container basket. | Lift a loaded basket and transport it to a designated target zone on the tablecloth. |
| **# demonstrations** | 100 | 100 | 100 | 100 |
| **# rollouts** | 50 | 50 | 50 | 50 |
| **Performance (%)** | 70 | 74 | 72 | 64 |

# D. Case Study

In this section, we present qualitative Real-World examples of BehaviorVLA. Empowered by VBE and PBD, BehaviorVLA exhibits superior performance on the *Generalization Tasks* (Figs. 8) as well as on the *Long-horizon Tasks* (Figs. 9).

*Table 7.* Overview of *Long-horizon Tasks* on Real-World.

| | **Long-Horizon Tasks** | | | |
|---|---|---|---|---|
| **Task name** | move and stack blocks on center | place containers on plate | pick and place blocks in bowl | place bottles and cans in basket |
| **Description** | Pick up blocks and place on the center of the table, then stack the red block onto the purple block. | Pick up containers from different locations and stably arrange them onto the plate. | Identify two spatially separated red blocks and execute a multi-stage pick-and-place sequence to deposit them both into the bowl. | Grasp multiple bottles and cans and transporting them into a large storage basket. |
| **# demonstrations** | 200 | 150 | 150 | 200 |
| **# rollouts** | 50 | 50 | 50 | 50 |
| **Performance (%)** | 58 | 54 | 60 | 48 |

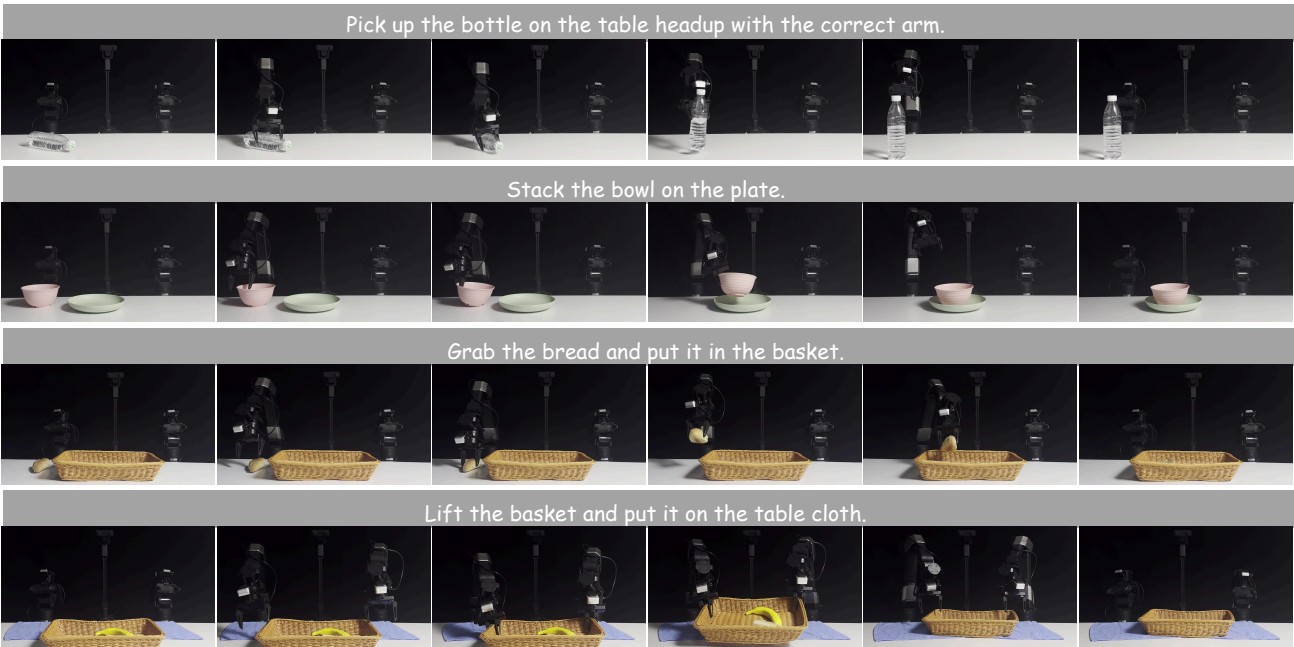

*Figure 8.* Qualitative results of BehaviorVLA on Real-World. From top to bottom, we illustrate four *Generalization Tasks*: *Adjust bottle*, *Stack bowl on plate*, *Place bread in basket*, and *Place basket on tablecloth*.The model demonstrates robust adaptability in scenarios requiring precise interaction, confirming the effectiveness of the learned visuomotor behavior manifold.

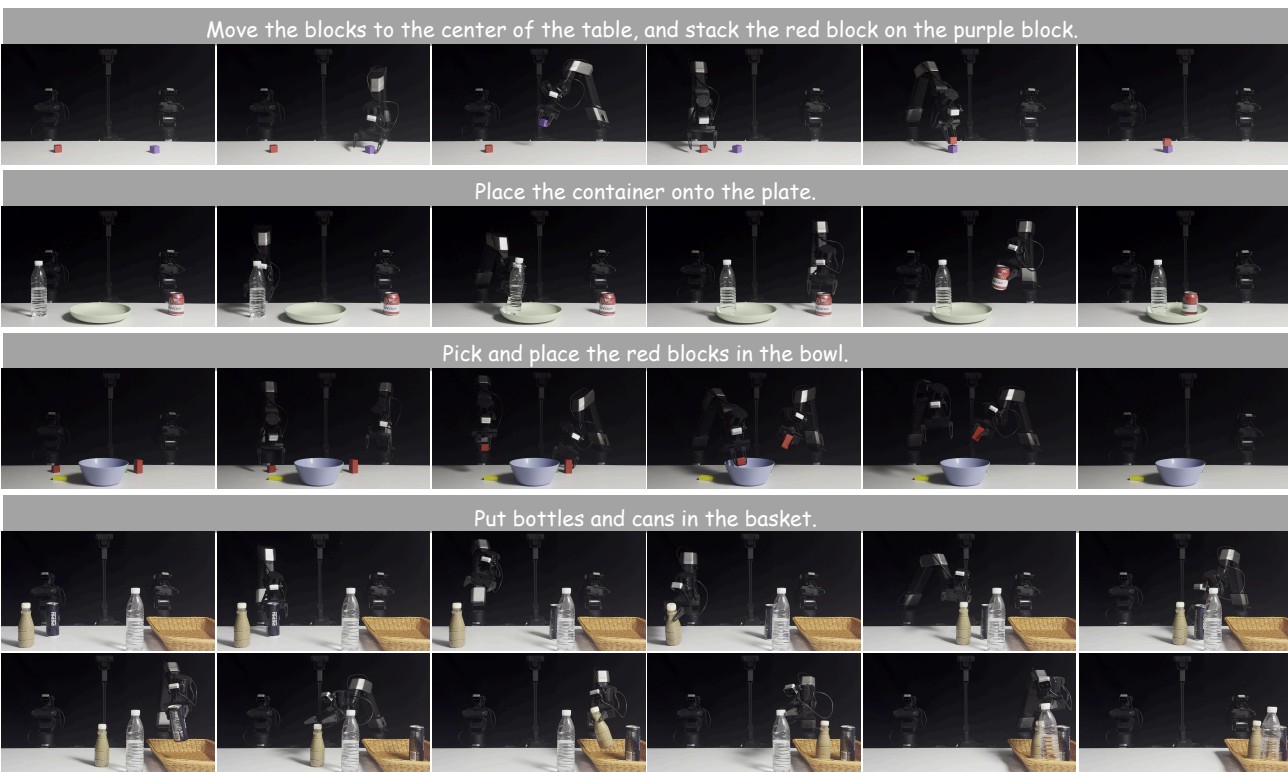

*Figure 9.* Qualitative results of BehaviorVLA on Real-World. From top to bottom, we illustrate four *Long-horizon Tasks*: *Move and stack blocks on center*, *Place containers on plate*, *Pick and place blocks in bowl*, and *Place bottles and cans in basket*.By conditioning the policy on a global prototype for structural guidance and dynamically tracking execution via phase variables, BehaviorVLA mitigates temporal drift, ensuring that local actions remain synchronized with the overall task progress.

