# OpenReview forum: "From Abstraction to Instantiation: Learning Behavioral Representation for Vision-Language-Action Model"
_ICML.cc/2026/Conference — ICML 2026 spotlight_

### Official Review · Reviewer_wDyF · 2026-02-22

**Soundness:** 3
**Presentation:** 3
**Significance:** 3
**Originality:** 2
**Overall Recommendation:** 5
**Confidence:** 4

**Summary:**

To address the performance degradation of Vision-Language-Action (VLA) models due to temporal fragmentation and static alignment when facing environmental distribution shifts, the authors propose the BehaviorVLA framework, which achieves robust robot control through an "abstraction-to-instantiation" mechanism.

**Compliance With Llm Reviewing Policy:**

Affirmed.

**Final Justification:**

Although I initially had some doubts and concerns about the authors' work, they responded very well to these questions during the rebuttal and supplemented the experiments accordingly. I now believe that the authors' work is ready for acceptance after revisions, so I am raising the score to 5.

**Key Questions For Authors:**

I have four questions here:

1. The paper's appendix states that iterative solutions based on flow matching introduce higher inference latency. So, what is the actual control frequency (Hz) of the Behavior VLA in real-world robot deployments? Does this latency hinder the model's application in contact-based manipulation tasks requiring high-frequency responses?

2. Since the global prototype relies heavily on the offline-built memory library, is its safe rollback/error reporting mechanism truly effective when the model encounters a new task that goes beyond the training manifold? Have the authors fully considered this and implemented corresponding strategies?

3. Ablation experiments indicate the existence of an optimal guiding strength coefficient to balance global structure and local control. Is this optimal value universal across all simulation benchmarks (RoboTwin 2.0, LIBERO, CALVIN) and real-world tasks, or must it be manually fine-tuned for each specific task? What is the actual robustness of the optimal value?

4. The global prototype is obtained by calculating the similarity between the query and all prototypes in memory, and then using a pooling method to weight the top K candidate prototypes. However, if the model is scaled to massive robot datasets on the internet, will this O(N) complexity retrieval mechanism become a serious computational and memory bottleneck, thus significantly degrading the model's performance? Have the authors adequately considered this?

**Limitations:**

yes

**Strengths And Weaknesses:**

The authors first propose a Visual Motion Behavior Encoder (VBE), which uses a three-stream architecture based on causal Mamba to extract long-period trajectories into a unified and noise-resistant behavioral representation. Secondly, they propose a Phase Conditional Behavior Decoder (PBD), which uses a "predictor-corrector" paradigm and dynamic phase states to strictly align the above representation with the real-time execution progress and transform it into precise actions, effectively overcoming time drift.

I believe the main weaknesses of this paper are as follows:

1. The proposed specific-to-general approach heavily relies on an offline prototype memory built from the training data. As the authors frankly admit in the appendix, this inherently limits the model's ability to adapt to novel, out-of-distribution task topologies, where the retrieved skeleton structures may exhibit poor stationarity. Compared to a simple regression baseline, the flow matching corrector relies on solving iterative differential equations, which introduces higher inference latency. This trade-off may pose a challenge for deployment on computationally limited platforms requiring high-frequency control, and this issue requires further discussion.

2. The mathematical formulas in Section 3.3 (Phase Conditional Behavior Decoder) are somewhat obscure. The complex transitions between latent anchor box generation, progress attention mechanisms, and continuous flow matching require more intuitive and detailed textual explanations to ensure the reproducibility of the results.

3. Its impact is relatively concentrated in the domain of robot manipulation and Visual Learning Array (VLA) models. The core mechanisms, particularly the three-stream (vision, action, behavior) Mamba encoder, are highly designed for sequences of physical action states and may be difficult to generalize to broader machine learning applications beyond embodied AI.

4. While the architecture synthesis is very effective, the individual components (e.g., Mamba for sequence modeling, conditional flow matching for generating policies, and behavior learning for retrieval enhancement) are all existing concepts. Its innovation lies primarily in building a collaborative pipeline rather than introducing entirely new mathematical or theoretical paradigms.

I would give a higher score if the necessity of the work could be better demonstrated.

---

> ### Author Rebuttal · Authors · 2026-03-31
>
> Thank you for your thoughtful review and constructive suggestions. We have responded to your concerns as follows. We would greatly appreciate it if you could consider giving us a higher rating.
> ### W1&Q1: Limitations of Offline Memory and Inference Latency in Real-World
>
> >R1: *Regarding OOD Generalization and Offline Memory*, we conducted Cross-Dataset in response to **wZNx R3** and data scarcity experiments in response to **Zcni R4**, indicating that the extracted prototypes capture domain-invariant structures, which helps maintain generalization under OOD shifts with limited data.
> >
> > *Regarding Inference Latency*, we conducted Inference Latency experiments in response to **Zcni R3**, showing that our framework operates at 155 Hz; the 27ms overhead does not hinder real-time control, while supporting improved success rates in contact-rich tasks.
> ### W2: Clarity and Reproducibility of the PBD Formulation
> >R2: In the revised manuscript, we will expand the textual explanations and provide a more intuitive, step-by-step algorithmic breakdown of these processes. We appreciate the valuable suggestion, which will enhance the readability and reproducibility of our work.
> ### W3: Limited Generalizability Beyond EAI
> >R3: While our primary focus and empirical validation are indeed within EAI, we believe the core ML paradigm holds potential beyond robotics. In our future work, we will explore applying this manifold learning paradigm to fields such as autonomous driving and game agents.
> ### W4: Innovation as a Principled Integration Rather Than a New Primitive
>
> >R4: - **Algorithmic Adaptation in VBE:** We leverage Mamba's input-dependent SSM filtering to discard high-frequency environmental noise. This adaptation is architecturally necessary: VBE-Mamba achieves a 62.5% SR, outperforming a standard Transformer baseline 54.5% as ablation in response to **EuXM R1** .
> > - **Constraining the Flow Policy:** Standard flow matching often causes high-frequency jitter in contact-rich tasks. PBD solves this not by simply adding another module, but by organically injecting the extracted behavior prototype as a structural prior into the flow's vector field. This constrains the generative policy to fit residual local dynamics.
> > - **Inseparable Integration:** VBE and PBD form a tightly coupled control loop. Removing VBE or PBD drops the real-world success rate by 16.7% and 9.6% respectively. This proves they function as an interdependent system rather than a sequentially stitched pipeline.
> ### Q2: Concern about Safe Rollback Mechanism
> >R5: Currently, our safety strategy is implicitly governed by the guidance strength ($\lambda$). As shown in the ablation response to **EuXM R5**, a moderate $\lambda$ (e.g., 0.5) acts as a "soft rollback," allowing the flow corrector's local dynamics to override flawed global prototypes and avoiding the performance drop caused by over-constraint. In future work, we plan to implement a confidence-based mechanism. When encountering novel tasks beyond the manifold, the system will dynamically adjust $\lambda$ based on uncertainty to ensure execution safety.
> ### Q3: Robustness and Universality of the Guidance Strength $\lambda$
>
> >R6: We added an ablation study sweeping $\lambda$ from 0.3 to 1.0 across CALVIN, and real-world tasks. The results show that $\lambda = 0.5$ consistently achieves peak or near-peak performance across nearly simulation and real-world environments, confirming the value's high universality.
> > |$\lambda$|CALVIN(Avg.Len)|Real-World(SR)|
> > |-|-|-|
> > |0.3|4.29|64%|
> > |0.5|4.36|70%|
> > |0.7|4.24|68%|
> > |1|4.20|58%|
> ### Q4: Scalability of Memory Retrieval Under Large-Scale Prototype Banks
>
> >R7: We address this scalability concern from three perspectives:
> > - **Single Retrieval per Task**: Retrieval of the global prototype $z_{\mathfrak{proto}}$ occurs only once at the beginning of a task. Subsequent real-time control is managed by the local phase $z_{phase}$, ensuring that the $O(N)$ retrieval does not affect execution frequency.
> > - **Minimal Memory Footprint**: Each prototypes are stored as compact latent vectors (e.g., dimension $D=256$). Even at an extreme scale, 10^6 prototypes occupy only 488.3 MiB of memory, ensuring the memory bank remains accessible for compute-constrained robotic platforms.
> > - **Scalability and Efficiency**: Ablation studies on large-scale memory banks in response to **Zcni R5** demonstrate that retrieval latency remains extremely low with 1,000 task categories containing multiple representations(e.g., 0.0042ms). The system maintains a robust Recall@5 at scale, confirming that the retrieval mechanism is both efficient and reliable as data increases.

---

> > ### Author Rebuttal · Reviewer_wDyF · 2026-04-01
> >
> > The author has already answered my questions quite well.

---

> > > ### Author Response · Authors · 2026-04-02
> > >
> > > Thank you very much for your thoughtful review and constructive comments. Your feedback has been very helpful in improving the quality and clarity of our paper. We are also very glad to know that our rebuttal has addressed all of your concerns.
> > >
> > > We also noticed your note in the original review indicating that the score could be increased if these concerns were resolved. If you feel that our responses have satisfactorily addressed the issues you raised, we would be sincerely grateful if you could kindly consider updating the score in your original review accordingly.

---

### Official Review · Reviewer_Zcni · 2026-03-05

**Soundness:** 3
**Presentation:** 3
**Significance:** 3
**Originality:** 3
**Overall Recommendation:** 5
**Confidence:** 4

**Summary:**

The authors present Behavior VLA to address the performance degradation of Vision-Language-Action (VLA) models under distribution shifts. The core argument is that standard VLA architectures struggle to learn generalized behavior representations, often suffering from short-horizon temporal fragmentation and static execution-alignment.To resolve this, the paper proposes a framework that decouples behavior into a global prototype ($z_{proto}$) and a real-time phase state ($z_{phase}$). This is achieved through two symmetric modules: the Visuomotor Behavior Encoder (VBE), which uses a three-stream causal Mamba architecture to aggregate long-horizon trajectories into a unified latent space ; and the Phase-conditioned Behavior Decoder (PBD), which acts as a Predictor-Corrector. The PBD unfolds structural priors aligned with the execution phase and injects them to guide a Conditional Flow Matching policy. The authors claim state-of-the-art success rates across RoboTwin 2.0, LIBERO, and CALVIN, and demonstrate that Behavior VLA matches the sim-to-real transfer performance of OpenVLA-OFT using only 50% of the required demonstration data.

**Compliance With Llm Reviewing Policy:**

Affirmed.

**Key Questions For Authors:**

Theoretical Justification of Prior Injection: In Eq. 11, you inject the structural prior additively into the noisy embedding space. Could you provide a more rigorous mathematical justification for why an additive shift effectively constrains the transport path of the vector field, rather than just biasing the initial condition?

Inference Latency Breakdown: The Predictor-Corrector paradigm requires iterative integration for the flow policy atop the VBE and PBD forward passes. Please provide a precise wall-clock latency analysis (in Hz) for the control loop on the real-world deployment hardware.

Memory Bank Scalability: How does the inference time and retrieval accuracy of $z_{proto}$ scale as the offline memory bank $\mathcal{M}$ grows from 50 tasks to thousands of diverse manipulation tasks? Does the attention mechanism in Eq. 6 suffer from catastrophic distraction?

Ablation on Data Scarcity: While 50% data efficiency is impressive relative to OpenVLA-OFT, how does Behavior VLA perform in a true low-data regime (e.g., 5-10 demonstrations per real-world task) where the manifold clustering in Phase 1 might fail?

**Limitations:**

The authors successfully included a Limitations appendix. They honestly identify that the specific-to-general abstraction is constrained by the topological diversity of the offline memory bank, which can cause the PBD to generate functionally incorrect actions if a novel task departs significantly from the learned manifold. They also rightly point out the inference latency issues inherent to solving iterative differential equations for flow matching on compute-constrained hardware. To strengthen this section, the authors should also explicitly discuss the risks of failure modes caused by the stochastic dropout mask ($m \sim Bernoulli(p)$) during inference out-of-distribution. Furthermore, an assessment of the visual encoder's susceptibility to adversarial lighting or viewpoint shifts that might corrupt the initial $z_{phase}$ mapping would make the limitations discussion complete.

**Strengths And Weaknesses:**

Strengths:

Originality: The explicit decoupling of behavior representations into a time-invariant global prototype $z_{proto}$ and a time-variant local phase $z_{phase}$, combined with a tri-stream Mamba architecture to filter environmental noise, is highly creative. This design elegantly breaks away from traditional action chunking and discrete latent code paradigms.

Quality: The empirical evaluation is exceptionally solid. The model significantly outperforms existing baselines on highly challenging benchmarks such as RoboTwin 2.0 Hard (58% success rate) and LIBERO (98% success rate). Furthermore, the dual-stage training strategy is logically rigorous.

Significance: The method demonstrates remarkable data efficiency in real-world sim-to-real transfer, achieving parity with OpenVLA-OFT using only 50% of the demonstration data. This provides highly valuable insights for scaling up robotic manipulation models.

Weaknesses:

Soundness Issues: The injection of the structural prior relies on a simple additive operation: $h_{\sigma}=e(a_{\sigma})+m\cdot Proj_{ob}(\mu_{prior})$. Theoretically, this does not strictly constrain the transport path or covariance of the flow model. Additionally, the ablation study explicitly indicates that the system is highly sensitive to the guidance strength scalar $\lambda$.

Novelty Scrutiny: Although the VBE and PBD framework design is clever , the underlying mechanisms heavily rely on existing Mamba and Conditional Flow Matching architectures. The overall work leans towards an "engineering assembly of SOTA modules," lacking fundamental mathematical primitive innovation.

Clarity: The paper fails to quantify the inference latency and computational overhead associated with querying the offline memory bank and online unrolling of Mamba states. This is crucial for evaluating its practicality in high-frequency control systems.

Generalization: In the evaluation of real-world "Generalization Tasks," 100 expert demonstrations were still collected per task. In physical robotics, this stretches the definition of true "few-shot" transfer, leaving its zero-shot generalization capabilities in completely out-of-distribution (OOD) environments questionable.

---

> ### Author Rebuttal · Authors · 2026-03-31
>
> Thank you for the thoughtful and encouraging review. We have responded to your concerns as follows. We would greatly appreciate it if you could consider giving us a higher rating.
> ### W1&Q1: Theoretical Justification of Prior Injection.
> >R1: **Mathematical Justification:** Eq. (11) should be interpreted as a soft geometric bias, rather than a hard constraint on all transport paths or covariances. First, the additive term is not only applied to the initial condition: with $B=\lambda,\mathrm{Proj}(\mu_{\text{prior}})$, inference solves
> >$$\frac{da_\sigma}{d\sigma}=v_\theta(e(a_\sigma)+B,\sigma,\Phi(O_t,L),\hat z_{\text{proto}}),$$
> > so the prior perturbs the vector field itself at every integration step. A first-order expansion gives
> > $$v_\theta(e(a)+B)\approx v_\theta(e(a))+J_{v_\theta,e}B,$$
> > which implies a state-dependent drift toward prior-aligned directions throughout transport. Second, since $\hat z_{\text{proto}}$ provides a stable global guide and PBD predicts a phase-aligned prior, the shift regularizes the flow toward high-probability task topology while still leaving local correction to $v_\theta$.
> >
> > **The guidance strength scalar:** As in the response to **EuXM R5**, the ablation shows a stable high-performance regime around $0.3 \sim 0.7$, with the best trade-off at 0.5. The drop at $\lambda=1.0$ is expected: an over-strong prior suppresses the fine-grained reactive correction needed for precise interaction.
> ### W2: Novelty Scrutiny.
> >R2: - **VBE Design:** VBE leverages the SSM's input-dependent time filtering to discard high-frequency noise, combined with cross-attention for spatial multimodal fusion. This is architecturally necessary: ablation in response to **EuXM R1** shows VBE-Mamba achieves 62.5% SR, outperforming VBE-Transformer (54.5%).
> > - **Innovation in PBD:** Standard flow matching suffers from high-frequency jitter in contact-rich tasks. PBD addresses this via a rigorous Predictor-Corrector paradigm: it unfolds the extracted prototype into a phase-aligned structural prior, guiding the flow policy to solely fit residual local dynamics.
> > - **Inseparable Coupling:** VBE and PBD form an inseparable logical loop. Ablations confirm that removing VBE causes a 16% real-world SR drop, while removing PBD drops SR by 9.6%. This quantitatively proves a highly interdependent algorithmic framework.
> ### W3&Q2: Inference Latency Breakdown
> >R3: We compared the single-step latency and wall-clock frequency of BehaviorVLA against the $\pi_{0.5}$. BehaviorVLA achieves a 162ms inference latency and operates at a high frequency of 155 Hz, introducing a marginal 27ms overhead over the baseline. This proves the model satisfies real-time high-frequency control requirements while boosting the real-world SR (62.5% vs 49%).
> > |  | SR/% | Latency/ms | Wall-clock Latency  /Hz |
> > | --- | --- | --- | --- |
> > | pi0.5 | 49 | 135 | 185 |
> > | BehaviorVLA | 62.5 | 162 | 155 |
> ### W4&Q4: Ablation on Data Scarcity
> >R4: We conducted an ablation using 5%(5 demos), 10%, 20%, and 50% of the training data on Generalization Tasks. With only 10% data, BehaviorVLA maintains a 20% SR, outperforming $\pi_{0.5}$ (5%) and OpenVLA-OFT (0%). This demonstrates that VBE's manifold clustering remains functional under severe data scarcity.
> > | Data Scale | OpenVLA-OFT  | $\pi_{0.5}$ | BehaviorVLA |
> > | --- | --- | --- | --- |
> > | 5% | 0% | 1% | **8%** |
> > | 10% | 0% | 5% | **20%** |
> > | 20% | 5.5% | 16% | **33%** |
> > | 50% | 18% | 28.5% | **49.5%** |
> ### Q3: Memory Bank Scalability
> >R5: we scaled the memory bank from 50 to 1000 and measured retrieval latency and retrieval recall. For prototypes from 1,000 task categories, the retrieval latency is only 0.0042ms, supporting high-frequency control. While Recall@1 naturally decreases, the Recall@5 of 80.6% confirms that the attention mechanism successfully aggregates diverse valid prototypes without catastrophic distraction.
> > | Bank size | Recall@1/% | Recall@5/% | Retrieval Latency/ms |
> > | --- | --- | --- | --- |
> > | 50 | 97.1 | 99.2 | 0.0002 |
> > | 100 | 88.4 | 94.7 | 0.0004 |
> > | 500 | 77.9 | 88.4 | 0.0019 |
> > | 1000 | 70.1 | 80.6 | 0.0042 |

---

> > ### Author Rebuttal · Reviewer_Zcni · 2026-04-02
> >
> > I thank the authors for their feedback. I have no further questions.

---

> > > ### Author Response · Authors · 2026-04-02
> > >
> > > Thank you very much for your thoughtful review and constructive comments. Your feedback has been very helpful in improving the quality and clarity of our paper. We are also very glad to know that our rebuttal has addressed all of your concerns. We would greatly appreciate it if you could consider giving us a higher rating.

---

### Official Review · Reviewer_wZNx · 2026-03-08

**Soundness:** 2
**Presentation:** 3
**Significance:** 3
**Originality:** 3
**Overall Recommendation:** 5
**Confidence:** 3

**Summary:**

This paper addresses the critical challenge of severe performance degradation in Vision-Language-Action (VLA) models under distribution shift, and proposes BehaviorVLA, a novel framework for robust robotic manipulation. Unlike conventional VLA models that directly map observations and language instructions to action sequences, the authors develop a visuomotor behavior encoder for specific-to-general abstraction, which distills instance-level details into generalizable behavioral priors. They further design a phase-conditioned behavior decoder to achieve general-to-specific instantiation, mapping abstract representations to precise, execution-aware actions and enabling the disentanglement of global task structure and local execution dynamics. In addition, a tailored training strategy is introduced for the proposed framework, with experiments and ablation studies presented to validate its effectiveness.

**Compliance With Llm Reviewing Policy:**

Affirmed.

**Final Justification:**

Based on the author's detailed response, my concerns have been resolved. The author's commitment to open-sourcing the project has also greatly enhanced my confidence in this work.

**Key Questions For Authors:**

1.	Will this lead to the framework yielding favorable performance on seen scenarios, while suffering from notable performance degradation on unseen scenarios? In addition, please clarify how to rigorously validate the effectiveness of the proposed task prototype extraction method.
2.	For short and simple tasks, can the traditional Transformer achieve better performance than Mamba? Does Mamba require more training data relative to the Transformer?

**Limitations:**

yes

**Strengths And Weaknesses:**

Strengths:

1.	The paper is well-structured, and the proposed method is clear.

2.	This paper identifies the limitations of existing frameworks and proposes a framework to attempt to address the issues identified.

Weaknesses:

1.	This paper constructs an offline memory bank and obtains the final prototype via similarity-weighted pooling over the top-k prototypes, where the value of k is a fixed constant. This paper does not explicitly explain the method for selecting the k value in the experiments, nor does it clarify whether different k values will exert a significant impact on the experimental results.

2.	This paper does not characterize the time efficiency of the proposed model, with only a passing mention that numerical integration introduces higher inference latency. It fails to provide additional comprehensive quantitative analyses of time efficiency, and for instance, does not explain the total number of tasks that can be processed by different frameworks under an identical time budget, as well as the corresponding number of successfully completed tasks.

3.	The performance of the proposed model seems to be strongly correlated with the size of the training dataset. There are no cross experiments between datasets to verify the generalization ability of prototype extraction.

4.	In Figure 2, the derivation of Zproto relies on the visual encoding from O₀to Oₜ. Therefore, Zproto, like Zphase, should be a time-varying vector.

---

> ### Author Rebuttal · Authors · 2026-03-31
>
> Thank you for the thoughtful review and constructive suggestions. We have responded to your concerns as follows. We would greatly appreciate it if you could consider giving us a higher rating.
> ### W1: This paper does not explicitly explain the method for selecting the k value in the experiments.
> > R1: To clarify the impact of the hyperparameter $k$, we conducted an ablation study on LIBERO for $k \in \{1, 3, 5, 10\}$.
> >
> > | k | Spatial | Object | Goal | Long |
> > | --- | --- | --- | --- | --- |
> > | 1 | 98.2 | 98.0 | 97.6 | 93 |
> > | 3 | **99.2** | 99 | 98.4 | 93.8 |
> > | 5 | **99.2** | **99.4** | **98.8** | **94.6** |
> > | 10 | 98.6 | 98.8 | 98.2 | 94 |
> >
> > The results demonstrate that $k=5$ provides the optimal balance. Specifically, a small $k$ lacks sufficient behavioral diversity and is sensitive to individual query biases, whereas a large $k$ may introduce retrieval noise from less relevant prototype.
> >
> ### W2: This paper does not characterize the time efficiency of the proposed model.
> >R2: To quantify the effective throughput under a fixed time budget, we conducted a 30-minute evaluation on real-world tasks, comparing successful completions and total attempts. Experimental results show that although BehaviorVLA makes slightly fewer total attempts within the 30-minute, it yields a 22.2% increase in overall successful task completions (66 vs. 54), justifying the computational trade-off.
> >
> > |  | Adjust bottles | Stack bowl on plate | Place containers on plate | Place bottles and cans in basket |
> > | --- | --- | --- | --- | --- |
> > | $\pi_{0.5}$ | 17/35(48.7%) | 22/33(66.7%) | 9/26(34.6%) | 6/21(28.6%) |
> > | ours | 20/32(62.5%) | 22/29(79.3%) | 14/23(60.9%) | 10/18(55.5%) |
> ### W3: There are no cross experiments between datasets to verify the generalization ability of prototype extraction.
> >R3: **Dataset Size Dependency** As illustrated in Fig. 3, BehaviorVLA achieves strong performance without relying on massive datasets: it matches the fully fine-tuned OpenVLA-OFT using merely 50% of the demonstrations.
> >
> > **Cross-Dataset Generalization** To verify the cross-dataset generalization ability of the prototypes, we conducted a Cross-Domain (Sim-to-Real) experiment on real-world tasks where the memory is built from RoboTwin. BehaviorVLA successfully executes physical tasks using only simulation prototypes, confirming *the generalization ability of prototype extraction.*
> >
> > | Task | SR |
> > | --- | --- |
> > | Adjust Bottle | 54% |
> > | Place Bread Basket | 58% |
> > | Stack Blocks Two | 36% |
> > | Stack Bowls Two | 44% |
> ### W4: In Figure 2, the derivation of Zproto relies on the visual encoding from O₀ to Oₜ.
> > R4: Thank the reviewer for pointing out this detail, which enhances the rigor of our manuscript. As defined in Sec 3.2.2 in main text, $z_{proto}$ is a time-invariant global prototype retrieved once using the initial observation, and the sequence arrow in Fig2 was a drafting oversight. We will correct it in the revision.
>
> ### Q1: Will this lead to the framework yielding favorable performance on seen scenarios.
> >R5: **Generalization on Unseen Scenarios:** BehaviorVLA improves scene-level generalization under shared task structures, not unrestricted generalization, which validates by:
> >
> > - **CALVIN ($ABC \rightarrow D$):** 4.36 Avg. Len. in transfer to unseen environments.
> > - **RoboTwin 2.0 (Hard):** 58% SR under domain randomization (novel clutter/textures).
> > - **Real-World:** 70% SR on real-world generalization tasks with varying lighting, layouts, and object poses.
> >
> > **Effectiveness of Prototype Extraction:** We validate the effectiveness of prototype extraction from three complementary perspectives.
> >
> > - **System Ablation:** Removing the VBE drops real-world SR from 70.0% to 60.0% (Tab.3), proving its necessity against environmental noise.
> > - **Architectural Comparison:** As in response to **EuXM R1,** Mamba-based VBE (98%) outperforms LSTM (96.4%) and Transformer (97.2%).
> > - **Qualitative Clustering:** Fig 6's t-SNE shows prototypes form discriminative semantic clusters, while ablating vision/action streams causes severe aliasing.
> ### Q2: For short and simple tasks, can the traditional Transformer achieve better performance than Mamba?
> >R6: Using the same training data, VBE-Mamba outperforms VBE-Transformer in both simulation and real-world evaluation as the response to **EuXM R1**. On LIBERO, VBE-Mamba achieves higher SR (98.0% vs. 97.2%) with lower latency (75 ms vs. 90 ms). On real-world tasks, it also delivers both better success rate (62.5% vs. 54.5%) and lower latency (162 ms vs. 192 ms).
> >

---

> > ### Author Rebuttal · Reviewer_wZNx · 2026-04-03
> >
> > The authors addressed my previous concerns well. My only remaining question is whether you plan to release the code. Making the implementation publicly available would greatly improve the work's contribution, reproducibility, and credibility. A clear commitment to open-sourcing the code would positively impact my final evaluation.

---

> > > ### Author Response · Authors · 2026-04-03
> > >
> > > Thank you for your insightful review. We are very glad to see that our responses have fully addressed your previous concerns. Your insightful suggestions have been instrumental in significantly improving the quality, clarity, and completeness of our paper.
> > >
> > > We are committed to open-sourcing our work; in fact, we have already prepared the codebase and demonstration videos, and we will make them publicly available immediately upon the acceptance of this paper. We sincerely hope that this addresses your final question and that you might kindly consider updating your score to reflect this resolution.

---

### Official Review · Reviewer_EuXM · 2026-03-13

**Soundness:** 2
**Presentation:** 2
**Significance:** 2
**Originality:** 4
**Overall Recommendation:** 5
**Confidence:** 4

**Summary:**

This paper proposes BehaviorVLA, which augments a VLA with an explicit behavioral representation through a visuomotor behavior encoder and a phase-conditioned behavior decoder. The goal is to better preserve spatial-temporal structure and improve consistency in long-horizon manipulation. I think this is a promising direction, since many current VLAs still do not model the behavior manifold well and can become temporally inconsistent. However, the overall method feels overly complicated, and the paper does not sufficiently disentangle which parts of the design are actually responsible for the gains.

**Compliance With Llm Reviewing Policy:**

Affirmed.

**Final Justification:**

The rebuttal addressed all the weaknesses I pointed out with responses and experiments; the score is updated.

**Key Questions For Authors:**

Please address the weaknesses listed above.

**Limitations:**

Yes

**Strengths And Weaknesses:**

Strengths
1. The paper targets an important and real weakness of current VLAs: inconsistent behavior over time and weak use of history.
2. Learning an explicit behavioral representation is an interesting direction beyond purely action-centric latent modeling.
3. The empirical effort is substantial, and I appreciate that the authors ran a number of baseline comparisons.

Weaknesses
1. The method has many design choices, but the paper does not ablate them in enough detail to show what actually matters. The current ablations are too coarse; removing VBE or PBD as a whole is not enough to justify the full architecture. It is unclear whether a much simpler history-aware design could achieve similar gains.
2. The related work discussion on history-based policies is incomplete. Relevant prior work such as RPT, ICRT, RoboSSM, and MTIL should be discussed more carefully.
3. The paper’s generalization claims feel overstated. The reported “generalization” tasks look more like robustness to lighting, object position, or longer horizons than broader language or task generalization. (here I am referring to "A Taxonomy for Evaluating Generalist Robot Policies" for the axes of generalization)
4. The connection to π0.5 is not explained clearly enough. It is hard to tell which part of the model is kept from pi0.5, additionally there's quite a bit of re-wiring. The paper would benefit from a clearer explanation.
5. Some results are difficult to assess without uncertainty estimates (especially given the values in the ablation are very close to each other)
6. (nit) Figure 1 is not presented well; the different y-axis scales and lack of error bars make the gains look more convincing than they may be.

---

> ### Author Rebuttal · Authors · 2026-03-31
>
> Thank you for the thoughtful and constructive review. We have responded to your concerns as follows. We would greatly appreciate it if you could consider giving us a higher rating.
>
> ### W1: The current ablations are too coarse.
>
> > R1: To address the concern, we conducted a component-level ablation study replacing the Mamba backbone in the VBE with LSTM and Transformer as follows:
> >
> > |  |LIBERO SR/%|Sim Latency/ms|Real-World SR/%|Real-World Latency/ms|
> > |-|-|-|-|-|
> > |Base|96.9|55|49|135|
> > |VBE-LSTM|96.4|66|42|150|
> > |VBE-Transformer|97.2|90|54.5|192|
> > |VBE-Mamba|98.0|75|62.5|162|
> > - Limitations of simpler designs: LSTM struggles to capture the complex long-horizon dependencies required for real-world tasks (SR drops to 42%).
> > - The efficiency bottleneck: While the Transformer improves performance over LSTM, its quadratic complexity introduces significant inference latency (192ms), which hinders high-frequency reactive control in real world.
> > - Optimal trade-off: Our Mamba-based causal architecture captures long-horizon topological structure while maintaining a linear complexity $\mathcal{O}(L)$. It achieves the best balance of performance (62.5% SR) and efficiency (162ms).
> ### W2: Relevant prior work such as RPT, ICRT, RoboSSM, and MTIL should be discussed more carefully.
> >R2:  **History-based and in-context policies.** Recent robot policies increasingly exploit long-horizon history for action generation. RPT studies Transformer-based sensorimotor pre-training with masked prediction over image, state, and action tokens. ICRT casts in-context imitation as next-token prediction with a causal Transformer. MTIL uses Mamba to encode full trajectory history, while RoboSSM adopts state-space models for more efficient long-context imitation. In contrast, BehaviorVLA uses history not only as context for action prediction, but to derive a retrieved global prototype and an online phase state for structure-aware, progress-aligned flow control.
> >
> > The complete version will be added in the revision.
> >
> ### W3: The paper’s generalization claims feel overstated.
> > R3: Following STAR-Gen, a more accurate statement is that our experiments study generalization under specific changes around a base task, not broad language generalization or open-ended new-task generalization. In our real-world setup, the Generalization Tasks mainly test visual/environmental axes, such as lighting, scene layout, and object pose/position, while the Long-horizon Tasks mainly test stable execution over longer action sequences. Our claim will be narrower in the revision: BehaviorVLA improves robustness on the axes we actually test, and it reduces temporal drift in long-horizon control.
> >
> ### W4: The connection to π0.5 is not explained clearly enough.
> >R4: We categorize the structural changes as follows:
> >
> > - What is kept: the vision-language backbone and the flow matching action policy.
> > - What is added: the VBE, PBD and the Behavior Memory Bank.
> > - "Re-wiring" Process: As shown in Fig. 2, backbone features are first used to retrieve a global prototype (z_{\text{proto}}), while VBE updates the online phase state (z_{\text{phase}}). PBD then fuses them to predict a structural prior, which is injected as a residual bias into the noisy action embedding of the original (\pi_{0.5}) flow policy.
> >
> > We will clarify this more explicitly in the revised manuscript.
> ### W5: Some results are difficult to assess without uncertainty estimates.
>
> >R5: To address the lack of uncertainty estimates, we evaluated LIBERO ablations across 3 random seeds.
> >
> > **Module Ablation:** LIBERO is an in-domain benchmark where the strong baseline $\pi_{0.5}$ already exceeds 96.9%, leaving limited headroom. While the added variance confirms our gains are statistically stable, the true necessity of VBE/PBD lies in OOD real-world tasks (Table 3 in main text), where they prevent severe performance drops (e.g., 70.0 to 57.0).
> >
> > |Method|Spatial|Object|Goal|Long|
> > |-|-|-|-|-|
> > |$\pi_{0.5}$|98.3±0.4|98.1±0.3|97.5±0.5|92.0±0.5|
> > |+VBE|98.4±0.3|98.7±0.4|98.4±0.4|93.1±0.6|
> > |+PBD|98.7±0.3|98.3±0.6|97.8±0.4|93.3±0.5|
> > |ours|**99.2±0.2**|**99.1±0.3**|**98.4±0.5**|**94.3±0.3**|
> >
> > **Parameter Ablation:** $\lambda=0.5$ provides the balance: lower values lack sufficient guidance for consistent trajectories, while higher values over-constrain the flow policy.
> >
> > |$\lambda$|Spatial|Object|Goal|Long|
> > |-|-|-|-|-|
> > |0.3|98.6±0.2|98.4±0.4|**98.4±0.5**|93.3±0.6|
> > |0.5|**99.2±0.2**|**99.1±0.3**|98.2±0.4|**94.3±0.3**|
> > |0.7|98.1±0.4|98.1±0.3|96.9±0.5|92.2±0.7|
> > |1.0|97.1±0.4|97.7±0.5|96.7 ± 0.3|90.5±0.6|
> ### W6: Figure 1 is not presented well
>
> >R6: In the revision, we will standardize the y-axis scales across panels and add error bars (or confidence intervals where applicable) to make the comparison more transparent. We appreciate this suggestion, which will make the figure clearer and the paper’s presentation more rigorous.

---

> > ### Author Rebuttal · Reviewer_EuXM · 2026-04-03
> >
> > I thank the authors for their feedback. I have no further questions.

---

> > > ### Author Response · Authors · 2026-04-03
> > >
> > > Thank you very much for your thoughtful review and constructive comments. Your feedback has been very helpful in improving the quality and clarity of our paper. We will carefully incorporate all of your comments into the revised version. We greatly appreciate your positive reassessment of our work and your increased score.

---

### Decision · Program_Chairs · 2026-04-30

**Decision:**

Accept (spotlight)

**Comment:**

This paper proposes BehaviorVLA, introducing explicit behavioral representations for VLA models via a global prototype (VBE) and phase-conditioned decoding (PBD). Reviewers agree the work addresses an important limitation (temporal inconsistency) and demonstrates very strong empirical performance, including improved robustness and data efficiency in sim-to-real settings.

The authors’ rebuttal successfully addressed the concerns from all reviewers. All reviewers converged to accept scores. AC and SAC concur with the reviewers and recommends acceptance.